# On Generalization Error Bounds of Noisy Gradient Methods for Non-Convex Learning

**Jian Li** [*]
Tsinghua University

**Xuanyuan Luo** [†]
Tsinghua University

**Mingda Qiao** [‡]
Stanford University

## Abstract

Generalization error (also known as the out-of-sample error) measures how well the hypothesis learned from training data generalizes to previously unseen data. Proving tight generalization error bounds is a central question in statistical learning theory. In this paper, we obtain generalization error bounds for learning general non-convex objectives, which has attracted significant attention in recent years. We develop a new framework, termed *Bayes-Stability*, for proving *algorithm-dependent* generalization error bounds. The new framework combines ideas from both the PAC-Bayesian theory and the notion of algorithmic stability. Applying the Bayes-Stability method, we obtain new data-dependent generalization bounds for stochastic gradient Langevin dynamics (SGLD) and several other noisy gradient methods (e.g., with momentum, mini-batch and acceleration, Entropy-SGD). Our result recovers (and is typically tighter than) a recent result in Mou et al. (2018) and improves upon the results in Pensia et al. (2018). Our experiments demonstrate that our data-dependent bounds can distinguish randomly labelled data from normal data, which provides an explanation to the intriguing phenomena observed in Zhang et al. (2017a). We also study the setting where the total loss is the sum of a bounded loss and an additional $\ell_2$ regularization term. We obtain new generalization bounds for the continuous Langevin dynamic in this setting by developing a new Log-Sobolev inequality for the parameter distribution at any time. Our new bounds are more desirable when the noise level of the process is not very small, and do not become vacuous even when $T$ tends to infinity.

## 1 Introduction

Non-convex stochastic optimization is the major workhorse of modern machine learning. For instance, the standard supervised learning on a model class parametrized by $\mathbb{R}^d$ can be formulated as the following optimization problem:

$$\min_{w \in \mathbb{R}^d} \mathbb{E}_{z \sim \mathcal{D}} [F(w, z)],$$

where $w$ denotes the model parameter, $\mathcal{D}$ is an unknown data distribution over the instance space $\mathcal{Z}$, and $F : \mathbb{R}^d \times \mathcal{Z} \to \mathbb{R}$ is a given objective function which may be non-convex. A learning algorithm takes as input a sequence $S = (z_1, z_2, \ldots, z_n)$ of $n$ data points sampled i.i.d. from $\mathcal{D}$, and outputs a (possibly randomized) parameter configuration $\hat{w} \in \mathbb{R}^d$.

A fundamental problem in learning theory is to understand the *generalization performance* of learning algorithms—is the algorithm guaranteed to output a model that generalizes well to the data distribution $\mathcal{D}$? Specifically, we aim to prove upper bounds on the *generalization error* $\mathrm{err}_{\mathrm{gen}}(S) = \mathcal{L}(\hat{w}, \mathcal{D}) - \mathcal{L}(\hat{w}, S)$, where $\mathcal{L}(\hat{w}, \mathcal{D}) = \mathbb{E}_{z \sim \mathcal{D}}[\mathcal{L}(\hat{w}, z)]$ and $\mathcal{L}(\hat{w}, S) = \frac{1}{n} \sum_{i=1}^{n} \mathcal{L}(\hat{w}, z_i)$ are the population and empirical losses, respectively. We note that the loss function $\mathcal{L}$ (e.g., the 0/1 loss) could be different from the objective function $F$ (e.g., the cross-entropy loss) used in the training process (which serves as a surrogate for the loss $\mathcal{L}$).

[*]lijian83@mail.tsinghua.edu.cn

[†]luo-xy19@mails.tsinghua.edu.cn

[‡]mqiao@stanford.edu

Classical learning theory relates the generalization error to various complexity measures (e.g., the VC-dimension and Rademacher complexity) of the model class. Directly applying these classical complexity measures, however, often fails to explain the recent success of over-parametrized neural networks, where the model complexity significantly exceeds the amount of available training data (see e.g., Zhang et al. (2017a)). By incorporating certain data-dependent quantities such as margin and compressibility into the classical framework, some recent work (e.g., Bartlett et al. (2017); Arora et al. (2018); Wei & Ma (2019)) obtains more meaningful generalization bounds in the deep learning context.

An alternative approach to generalization is to prove *algorithm-dependent* bounds. One celebrated example along this line is the algorithmic stability framework initiated by Bousquet & Elisseeff (2002). Roughly speaking, the generalization error can be bounded by the stability of the algorithm (see Section 2 for the details). Using this framework, Hardt et al. (2016) study the stability (hence the generalization) of stochastic gradient descent (SGD) for both convex and non-convex functions. Their work motivates recent study of the generalization performance of several other gradient-based optimization methods: Kuzborskij & Lampert (2018); London (2016); Chaudhari et al. (2017); Raginsky et al. (2017); Mou et al. (2018); Pensia et al. (2018); Chen et al. (2018).

In this paper, we study the algorithmic stability and generalization performance of various iterative gradient-based method, with certain continuous noise injected in each iteration, in a non-convex setting. As a concrete example, we consider the stochastic gradient Langevin dynamics (SGLD) (see Raginsky et al. (2017); Mou et al. (2018); Pensia et al. (2018)). Viewed as a variant of SGD, SGLD adds an isotropic Gaussian noise at every update step:

$$W_t \leftarrow W_{t-1} - \gamma_t g_t(W_{t-1}) + \frac{\sigma_t}{\sqrt{2}} \mathcal{N}(0, I_d), \tag{1}$$

where $g_t(W_{t-1})$ denotes either the full gradient or the gradient over a mini-batch sampled from training dataset. We also study a continuous version of (1), which is the dynamic defined by the following stochastic differential equation (SDE):

$$dW_t = -\nabla F(W_t)\, dt + \sqrt{2\beta^{-1}}\, dB_t, \tag{2}$$

where $B_t$ is the standard Brownian motion.

## 1.1 RELATED WORK

Most related to our work is the study of algorithm-dependent generalization bounds of stochastic gradient methods. Hardt et al. (2016) first study the generalization performance of SGD via algorithmic stability. They prove a generalization bound that scales linearly with $T$, the number of iterations, when the loss function is convex, but their results for general non-convex optimization are more restricted. London (2017) and Rivasplata et al. (2018) also combine ideas from both PAC-Bayesian and algorithm stability. However, these works are essentially different from ours. In London (2017), the prior and posterior are distributions on the hyperparameter space instead of distributions on the hypothesis space. Rivasplata et al. (2018) study the hypothesis stability measured by the distance on the hypothesis space in a setting where the returned hypothesis (model parameter) is perturbed by a Gaussian noise. Our work is a follow-up of the recent work by Mou et al. (2018), in which they provide generalization bounds for SGLD from both stability and PAC-Bayesian perspectives. Another closely related work by Pensia et al. (2018) derives similar bounds for noisy stochastic gradient methods, based on the information theoretic framework of Xu & Raginsky (2017). However, their bounds scale as $O(\sqrt{T/n})$ ($n$ is the size of the training dataset) and are sub-optimal even for SGLD.

We acknowledge that besides the algorithm-dependent approach that we follow, recent advances in learning theory aim to explain the generalization performance of neural networks from many other perspectives. Some of the most prominent ideas include bounding the network capacity by the norms of weight matrices Neyshabur et al. (2015); Liang et al. (2019), margin theory Bartlett et al. (2017); Wei et al. (2019), PAC-Bayesian theory Dziugaite & Roy (2017); Neyshabur et al. (2018); Dziugaite & Roy (2018), network compressibility Arora et al. (2018), and over-parametrization Du et al. (2019); Allen-Zhu et al. (2019); Zou et al. (2018); Chizat et al. (2019). Most of these results are stated in the context of neural networks (some are tailored to networks with specific architecture), whereas our work addresses generalization in non-convex stochastic optimization in general. We

also note that some recent work provides explanations for the phenomenon reported in Zhang et al. (2017a) from a variety of different perspectives (e.g., Bartlett et al. (2017); Arora et al. (2018; 2019)).

Welling & Teh (2011) first consider stochastic gradient Langevin dynamics (SGLD) as a sampling algorithm in the Bayesian inference context. Raginsky et al. (2017) give a non-asymptotic analysis and establish the finite-time convergence guarantee of SGLD to an approximate global minimum. Zhang et al. (2017b) analyze the hitting time of SGLD and prove that SGLD converges to an approximate local minimum. These results are further improved and generalized to a family of Langevin dynamics based algorithms by the subsequent work of Xu et al. (2018).

## 1.2 OVERVIEW OF OUR RESULTS

In this paper, we provide generalization guarantees for the noisy variants of several popular stochastic gradient methods.

**The Bayes-Stability method and data-dependent generalization bounds.** We develop a new method for proving generalization bounds, termed as Bayes-Stability, by incorporating ideas from the PAC-Bayesian theory into the stability framework. In particular, assuming the loss takes value in $[0, C]$, our method shows that the generalization error is bounded by both $2C \, \mathbb{E}_z[\sqrt{2\mathrm{KL}(P, Q_z)}]$ and $2C \, \mathbb{E}_z[\sqrt{2\mathrm{KL}(Q_z, P)}]$, where $P$ is a prior distribution independent of the training set $S$, and $Q_z$ is the expected posterior distribution conditioned on $z_n = z$ (i.e., the last training data is $z$). The formal definition and the results can be found in Definition 5 and Theorem 7.

Inspired by Lever et al. (2013), instead of using a fixed prior distribution, we bound the KL-divergence from the posterior to a *distribution-dependent* prior. This enables us to derive the following generalization error bound that depends on the expected norm of the gradient along the optimization path:

$$\mathrm{err}_{\mathrm{gen}} = O\left( \frac{C}{n} \sqrt{ \mathbb{E}_{S}\left[ \sum_{t=1}^{T} \frac{\gamma_t^2}{\sigma_t^2} \mathbf{g}_{\mathrm{e}}(t) \right] } \right). \tag{3}$$

Here $S$ is the dataset and $\mathbf{g}_{\mathrm{e}}(t) = \mathbb{E}_{W_{t-1}}[\frac{1}{n}\sum_{i=1}^{n} \|\nabla F(W_{t-1}, z_i)\|^2]$ is the expected empirical squared gradient norm at step $t$; see Theorem 11 for the details.

Compared with the previous $O\left(\frac{LC}{n}\sqrt{\sum_t \frac{\gamma_t^2}{\sigma_t^2}}\right)$ bound in (Mou et al., 2018, Theorem 1), where $L$ is the global Lipschitz constant of the loss, our new bound (3) depends on the data distribution and is typically tighter (as the gradient norm is at most $L$). In modern deep neural networks, the worst-case Lipschitz constant $L$ can be quite large, and typically much larger than the expected empirical gradient norm along the optimization trajectory. Specifically, in the later stage of the training, the expected empirical gradient is small (see Figure 1(d) for the details). Hence, our generalization bound does not grow much even if we train longer at this stage.

Our new bound also offers an explanation to the difference between training on correct and random labels observed by Zhang et al. (2017a). In particular, we show empirically that the sum of expected squared gradient norm (along the optimization path) is significantly higher when the training labels are replaced with random labels (Section 3.1, Figure 1, Appendix C.2).

We would also like to mention the PAC-Bayesian bound (for SGLD with $\ell_2$-regularization) proposed by Mou et al. (2018). (This bound is different from what we mentioned before; see Theorem 2 in their paper.) Their bound scales as $O(1/\sqrt{n})$ and the numerator of their bound has a similar sum of gradient norms (with a decaying weight if the regularization coefficient $\lambda > 0$). Their bound is based on the PAC-Bayesian approach and holds with high probability, while our bound only holds in expectation.

**Extensions.** We remark that our technique allows for an arguably simpler proof of (Mou et al., 2018, Theorem 1); the original proof is based on SDE and Fokker-Planck equation. More importantly, our technique can be easily extended to handle mini-batches and a variety of general settings as follows.

1. **Extension to other gradient-based methods.** Our results naturally extends to other noisy stochastic gradient methods including momentum due to Polyak (1964) (Theorem 26), Nes-

terov's accelerated gradient method in Nesterov (1983) (Theorem 26), and Entropy-SGD proposed by Chaudhari et al. (2017) (Theorem 27).

2. **Extension to general noises.** The proof of the generalization bound in Mou et al. (2018) relies heavily on that the noise is Gaussian[1], which makes it difficult to generalize to other noise distributions such as the Laplace distribution. In contrast, our analysis easily carries over to the class of log-Lipschitz noises (i.e., noises drawn from distributions with Lipschitz log densities).

3. **Pathwise stability.** In practice, it is also natural to output a certain function of the entire optimization path, e.g., the one with the smallest empirical risk or a weighted average. We show that the same generalization bound holds for all such variants (Remark 12). We note that the analysis in an independent work of Pensia et al. (2018) also satisfies this property, yet their bound is $O\left(\sqrt{C^2 L^2 n^{-1} \sum_{t=1}^{T} \eta_t^2 / \sigma_t^2}\right)$ (see Corollary 1 in their work), which scales at a slower $O(1/\sqrt{n})$ rate (instead of $O(1/n)$) when dealing with $C$-bounded loss.[2]

**Generalization bounds with $\ell_2$ regularization via Log-Sobolev inequalities.** We also study the setting where the total objective function $F$ is the sum of a $C$-bounded differentiable objective $F_0$ and an additional $\ell_2$ regularization term $\frac{\lambda}{2} \|w\|_2^2$. In this case, $F$ can be treated as a perturbation of a quadratic function, and the continuous Langevin dynamics (CLD) is well understood for quadratic functions. We obtain two generalization bounds for CLD, both via the technique of Log-Sobolev inequalities, a powerful tool for proving the convergence rate of CLD. One of our bounds is as follows (Theorem 15):

$$\text{err}_{\text{gen}} \leq \frac{2e^{4\beta C} CL}{n} \sqrt{\frac{\beta}{\lambda} \left(1 - \exp\left(-\frac{\lambda T}{e^{8\beta C}}\right)\right)}. \tag{4}$$

The above bound has the following advantages:

1. Applying $e^{-x} \geq 1 - x$, one can see that our bound is at most $O(\sqrt{T}/n)$, which matches the previous bound in (Mou et al., 2018, Proposition 8)[3].

2. As time $T$ grows, the bound is upper bounded by and approaches to $2e^{4\beta C} CL n^{-1} \sqrt{\beta/\lambda}$ (unlike the previous $O(\sqrt{T}/n)$ bound that goes to infinity as $T \to +\infty$).

3. If the noise level is not so small (i.e., $\beta$ is not very large), the generalization bound is quite desirable.

Our analysis is based on a Log-Sobolev inequality (LSI) for the parameter distribution at time $t$, whereas most known LSIs only hold for the stationary distribution of the Markov process. We prove the new LSI by exploiting the variational formulation of the entropy formula.

## 2 PRELIMINARIES

**Notations.** We use $\mathcal{D}$ to denote the data distribution. The training dataset $S = (z_1, \ldots, z_n)$ is a sequence of $n$ independent samples drawn from $\mathcal{D}$. $S, S' \in \mathcal{Z}^n$ are called *neighboring datasets* if and only if they differ at exactly one data point (we could assume without loss of generality that $z_n \neq z'_n$). Let $F(w, z)$ and $\mathcal{L}(w, z)$ be the objective and the loss functions, respectively, where $w \in \mathbb{R}^d$ denotes a model parameter and $z \in \mathcal{Z}$ is a data point. Define $F(w, S) = \frac{1}{n} \sum_{i=1}^{n} F(w, z_i)$ and $F(w, \mathcal{D}) = \mathbb{E}_{z \sim \mathcal{D}}[F(w, z)]$; $\mathcal{L}(w, S)$ and $\mathcal{L}(w, \mathcal{D})$ are defined similarly. A learning algorithm $\mathcal{A}$ takes as input a dataset $S$, and outputs a parameter $w \in \mathbb{R}^d$ randomly. Let $G$ be the set of all possible mini-batches. $G_n = \{B \in G : n \in B\}$ denotes the collection of mini-batches that contain the $n$-th data point, while $\overline{G_n} = G \setminus G_n$. Let $\text{diam}(A) = \sup_{x,y \in A} \|x - y\|_2$ denote the diameter of a set $A$.

---

[1]In particular, their proof leverages the Fokker-Planck equation, which describes the time evolution of the density function associated with the Langevin dynamics and can only handle Gaussian noise.

[2]They assume the loss is sub-Gaussian. By Hoeffding's lemma, $C$-bounded random variables are sub-Gaussian with parameter $C$.

[3]The proof of their $O(\sqrt{T}/n)$ bound can be easily extended to our setting with $\ell_2$ regularization.

**Definition 1** (*L*-lipschitz). *A function $F : \mathbb{R}^d \times \mathcal{Z} \to \mathbb{R}$ is L-lipschitz if and only if $|F(w_1, z) - F(w_2, z)| \leq L \|w_1 - w_2\|_2$ holds for any $w_1, w_2 \in \mathbb{R}^d$ and $z \in \mathcal{Z}$.*

**Definition 2** (Expected generalization error). *The expected generalization error of a learning algorithm $\mathcal{A}$ is defined as*

$$\text{err}_{gen} := \underset{S \sim \mathcal{D}^n}{\mathbb{E}}[\text{err}_{gen}(S)] = \underset{S \sim \mathcal{D}^n, \mathcal{A}}{\mathbb{E}}[\mathcal{L}(\mathcal{A}(S), \mathcal{D}) - \mathcal{L}(\mathcal{A}(S), S)].$$

**Algorithmic Stability.** Intuitively, a learning algorithm that is stable (i.e., a small perturbation of the training data does not affect its output too much) can generalize well. In the seminal work of Bousquet & Elisseeff (2002) (see also Hardt et al. (2016)), the authors formally defined algorithmic stability and established a close connection between the stability of a learning algorithm and its generalization performance.

**Definition 3** (Uniform stability). *(Bousquet & Elisseeff (2002); Elisseeff et al. (2005)) A randomized algorithm $\mathcal{A}$ is $\epsilon_n$-uniformly stable w.r.t. loss $\mathcal{L}$, if for all neighboring sets $S, S' \in \mathcal{Z}^n$, it holds that*

$$\sup_{z \in \mathcal{Z}} |\mathbb{E}_{\mathcal{A}}[\mathcal{L}(w_S, z)] - \mathbb{E}_{\mathcal{A}}[\mathcal{L}(w_{S'}, z)]| \leq \epsilon_n,$$

*where $w_S$ and $w_{S'}$ denote the outputs of $\mathcal{A}$ on $S$ and $S'$ respectively.*

**Lemma 4** (Generalization in expectation). *(Hardt et al. (2016)) Suppose a randomized algorithm $\mathcal{A}$ is $\epsilon_n$-uniformly stable. Then, $|\text{err}_{gen}| \leq \epsilon_n$.*

## 3 BAYES-STABILITY METHOD

In this section, we incorporate ideas from the PAC-Bayesian theory (see e.g., Lever et al. (2013)) into the algorithmic stability framework. Combined with the technical tools introduced in previous sections, the new framework enables us to prove tighter data-dependent generalization bounds.

First, we define the posterior of a dataset and the posterior of a single data point.

**Definition 5** (Single-point posterior). *Let $Q_S$ be the posterior distribution of the parameter for a given training dataset $S = (z_1, \ldots, z_n)$. In other words, it is the probability distribution of the output of the learning algorithm on dataset $S$ (e.g., for $T$ iterations of SGLD in (1), $Q_S$ is the pdf of $W_T$). The* **single-point posterior** $Q_{(i,z)}$ *is defined as*

$$Q_{(i,z)} = \underset{(z_1, \ldots, z_{i-1}, z_{i+1}, \ldots z_n)}{\mathbb{E}} \left[ Q_{(z_1, \ldots, z_{i-1}, z, z_{i+1}, \ldots, z_n)} \right].$$

For convenience, we make the following natural assumption on the learning algorithm:

**Assumption 6** (Order-independent). *For any fixed dataset $S = (z_1, \ldots, z_n)$ and any permutation $p$, $Q_S$ is the same as $Q_{S^p}$, where $S^p = (z_{p_1}, \ldots, z_{p_n})$.*

Assumption 6 implies $Q_{(1,z)} = \cdots = Q_{(n,z)}$, so we use $Q_z$ as a shorthand for $Q_{(i,z)}$ in the following. Note that this assumption can be easily satisfied by letting the learning algorithm randomly permute the training data at the beginning. It is also easy to verify that both SGD and SGLD satisfy the order-independent assumption.

Now, we state our new Bayes-stability framework, which holds for any prior distribution $P$ over the parameter space that is independent of the training dataset $S$.

**Theorem 7** (Bayes-Stability). *Suppose the loss function $\mathcal{L}(w, z)$ is $C$-bounded and the learning algorithm is order-independent (Assumption 6). Then for any prior distribution $P$ not depending on $S$, the generalization error is bounded by both $2C \, \mathbb{E}_z \left[ \sqrt{2\text{KL}(P, Q_z)} \right]$ and $2C \, \mathbb{E}_z \left[ \sqrt{2\text{KL}(Q_z, P)} \right]$.*

**Remark 8.** *Our Bayes-Stability framework originates from the algorithmic stability framework, and hence is similar to the notions of uniform stability and leave-one-out error (see Elisseeff et al. (2003)). However, there are important differences. Uniform stability is a distribution-independent property, while Bayes-Stability can incorporate the information of the data distribution (through the prior $P$). Leave-one-out error measures the loss of a learned model on an unseen data point, yet Bayes-Stability focuses on the extent to which a single data point affects the outcome of the learning algorithm (compared to the prior).*

To establish an intuition, we first apply this framework to obtain an expectation generalization bound for (full) gradient Langevin dynamics (GLD), which is a special case of SGLD in (1) (i.e., GLD uses the full gradient $\nabla_w F(W_{t-1}, S)$ as $g_t(W_{t-1})$).

**Theorem 9.** *Suppose that the loss function $\mathcal{L}$ is $C$-bounded. Then we have the following expected generalization bound for $T$ iterations of GLD:*

$$\mathrm{err}_{gen} \leq \frac{2\sqrt{2}C}{n} \sqrt{\underset{S\sim\mathcal{D}^n}{\mathbb{E}}\left[\sum_{t=1}^{T}\frac{\gamma_t^2}{\sigma_t^2}\mathbf{g}_\mathrm{e}(t)\right]},$$

*where $\mathbf{g}_\mathrm{e}(t) = \mathbb{E}_{w\sim W_{t-1}}[\frac{1}{n}\sum_{i=1}^{n}\|\nabla F(w, z_i)\|_2^2]$ is the empirical squared gradient norm, and $W_t$ is the parameter at step $t$ of GLD.*

**Proof** The proof builds upon the following technical lemma, which we prove in Appendix A.2.

**Lemma 10.** *Let $(W_0, \ldots, W_T)$ and $(W_0', \ldots, W_T')$ be two independent sequences of random variables such that for each $t \in \{0, \ldots, T\}$, $W_t$ and $W_t'$ have the same support. Suppose $W_0$ and $W_0'$ follow the same distribution. Then,*

$$\mathrm{KL}(W_{\leq T}, W_{\leq T}') = \sum_{t=1}^{T}\underset{w_{<t}\sim W_{<t}}{\mathbb{E}}[\mathrm{KL}(W_t|W_{<t} = w_{<t}, W_t'|W_{<t}' = w_{<t})],$$

*where $W_{\leq t}$ denotes $(W_0, \ldots, W_t)$ and $W_{<t}$ denotes $W_{\leq t-1}$.*

Define $P = \mathbb{E}_{\overline{S}\sim\mathcal{D}^{n-1}}[Q_{(\overline{S},\mathbf{0})}]$, where $\mathbf{0}$ denotes the zero data point (i.e., $F(w, \mathbf{0}) = 0$ for any $w$). Theorem 7 shows that

$$\mathrm{err}_{gen} \leq 2C\underset{z}{\mathbb{E}}\sqrt{2\mathrm{KL}(Q_z, P)}. \tag{5}$$

By the convexity of KL-divergence, for a fixed $z \in \mathcal{Z}$, we have

$$\mathrm{KL}(Q_z, P) = \mathrm{KL}\left(\underset{\overline{S}}{\mathbb{E}}[Q_{(\overline{S},z)}], \underset{\overline{S}}{\mathbb{E}}[Q_{(\overline{S},\mathbf{0})}]\right) \leq \underset{\overline{S}}{\mathbb{E}}\left[\mathrm{KL}\left(Q_{(\overline{S},z)}, Q_{(\overline{S},\mathbf{0})}\right)\right]. \tag{6}$$

Let $(W_t)_{t\geq 0}$ and $(W_t')_{t\geq 0}$ be the training process of GLD for $S = (\overline{S}, z)$ and $S' = (\overline{S}, \mathbf{0})$, respectively. Note that for a fixed $w_{<t}$, both $W_t|W_{<t} = w_{<t}$ and $W_t'|W_{<t}' = w_{<t}$ are Gaussian distributions. Since $\mathrm{KL}(\mathcal{N}(\mu_1, \sigma^2 I), \mathcal{N}(\mu_2, \sigma^2 I)) = \frac{\|\mu_1 - \mu_2\|_2^2}{2\sigma^2}$ (see Lemma 18 in Appendix A.2).

$$\mathrm{KL}(W_t|W_{<t} = w_{<t}, W_t'|W_{<t}' = w_{<t}) = \frac{\gamma_t^2\|\nabla F(w_{t-1}, z)\|_2^2}{\sigma_t^2 n^2}.$$

Applying Lemma 10 and $\mathrm{KL}(W_T, W_T') \leq \mathrm{KL}(W_{\leq T}, W_{\leq T}')$ gives

$$\mathrm{KL}(Q_S, Q_{S'}) \leq \frac{1}{n^2}\sum_{t=1}^{T}\frac{\gamma_t^2}{\sigma_t^2}\underset{w\sim W_{t-1}}{\mathbb{E}}\|\nabla F(w, z)\|_2^2.$$

Recall that $W_{t-1}$ is the parameter at step $t-1$ using $S = (\overline{S}, z)$ as dataset. In this case, we can rewrite $z$ as $z_n$ since it is the $n$-th data point of $S$. Note that SGLD satisfies the order-independent assumption, we can rewrite $z$ as $z_i$ for all $i \in [n]$. Together with (5), (6), and using $\frac{1}{n}\sum_{i=1}^{n}\sqrt{x_i} \leq \sqrt{\frac{1}{n}\sum_{i=1}^{n}x_i}$, we can prove this theorem. ∎

More generally, we give the following bound for SGLD. The proof is similar to that of Theorem 9; the difference is that we need to bound the KL-divergence between two Gaussian mixtures instead of two Gaussians. This proof is more technical and deferred to Appendix A.3.

**Theorem 11.** *Suppose that the loss function $\mathcal{L}$ is $C$-bounded and the objective function $f$ is $L$-lipschitz. Assume that the following conditions hold:*

    *1. Batch size $b \leq n/2$.*

2. *Learning rate $\gamma_t \leq \sigma_t/(20L)$.*

*Then, the following expected generalization error bound holds for $T$ iterations of SGLD* (1)*:*

$$\text{err}_{gen} \leq \frac{8.12C}{n} \sqrt{\mathop{\mathbb{E}}_{S \sim \mathcal{D}^n} \left[ \sum_{t=1}^{T} \frac{\gamma_t^2}{\sigma_t^2} \mathbf{g}_{\text{e}}(t) \right]}, \qquad \text{(empirical norm)}$$

*where $\mathbf{g}_{\text{e}}(t) = \mathbb{E}_{w \sim W_{t-1}}[\frac{1}{n}\sum_{i=1}^{n} \|\nabla F(w, z_i)\|_2^2]$ is the empirical squared gradient norm, and $W_t$ is the parameter at step $t$ of SGLD.*

Furthermore, based on essentially the same proof, we can obtain the following bound that depends on the *population gradient norm*:

$$\text{err}_{\text{gen}} \leq \frac{8.12C}{n} \sqrt{\mathop{\mathbb{E}}_{S'} \left[ \sum_{t=1}^{T} \frac{\gamma_t^2}{\sigma_t^2} \mathop{\mathbb{E}}_{w \sim W'_{t-1}} \left[ \mathop{\mathbb{E}}_{z \sim D} \|\nabla F(w, z)\|_2^2 \right] \right]}.$$

The full proofs of the above results are postponed to Appendix A, and we provide some remarks about the new bounds.

**Remark 12.** *In fact, our proof establishes that the above upper bound holds for the two sequences $W_{\leq T}$ and $W'_{\leq T}$: $\text{KL}(W_{\leq T}, W'_{\leq T}) \leq \frac{8.12}{n^2} \sum_{t=1}^{T} \frac{\gamma_t^2}{\sigma_t^2} \mathbf{g}_{\text{e}}(t)$. Hence, our bound holds for any sufficiently regular function over the parameter sequences: $\text{KL}(f(W_{\leq T}), f(W'_{\leq T})) \leq \frac{8.12}{n^2} \sum_{t=1}^{T} \frac{\gamma_t^2}{\sigma_t^2} \mathbf{g}_{\text{e}}(t)$. In particular, our generalization error bound automatically extends to several variants of SGLD, such as outputting the average of the trajectory, the average of the suffix of certain length, or the exponential moving average.*

**Remark 13** (High-probability bounds)**.** *By relaxing the expected squared gradient norm term to $L^2$ and using the uniform stability framework, our proof can be adapted to recover the $O(LC\sqrt{T}/n)$ bound in (Mou et al., 2018, Theorem 1). Then, we can apply the recent results of Feldman & Vondrak (2019) to provide a generalization error bound of $\tilde{O}(LC\sqrt{T}/n + 1/\sqrt{n})$ that holds with high probability. (Here $\tilde{O}$ hides poly-logarithmic factors.) When $T$ is at least linear in $n$, the additional $1/\sqrt{n}$ term is not dominating.*

### 3.1 EXPERIMENT

**Distinguish random from normal.** Inspired by Zhang et al. (2017a), we run both GLD (Figure 1) and SGLD (Appendix C.2) to fit both normal data and randomly labeled data (see Appendix C for more experiment details). As shown in Figure 1 and Figure 3 in Appendix C.2, a larger random label portion $p$ leads to both much higher generalization error and much larger generalization error bound. Moreover, the shapes of the curves of our bounds look quite similar to those of the generalization error curves.

Note that in (b) and (c) of Figure 1, the scales in the $y$-axis are different. We list some possible reasons that may explain why our bound is larger than the actual generalization error. (1) as we explained in Remark 12, our bounds (Theorem 9 and 11) hold for any trajectory-based output, and are much stronger than upper bounds for the last point on the trajectory. (2) The constant we can prove in Lemma 21 may not be very tight. (3) The variance of Gaussian noise $\sigma_t^2$ is not large enough in our experiment. However, if we choose a larger variance, fitting the random labeled training data becomes quite slow. Hence, we use a small data size ($n = 10000$) for the above reason. We also run an extra experiment for GLD on the full MNIST dataset ($n = 60000$) without label corruption (see Figure 2 in the Appendix C). We can see that our bound is *non-vacuous* (since GLD—which computes the full gradients—took a long time to converge, we stopped when we achieved 90% training accuracy). [4]

---

[4] We highlight another difficulty in proving non-vacuous generalization error bounds when the data are randomly labeled. Consider a 10-class classification setting where all the labels are random. For any sufficiently small data size, there is always a deep neural network that perfectly fits the dataset. Thus, the training error is zero while the population error is 90%. In this case, any valid generalization error bound should be larger than 0.9. Then, the theoretical bound would still be vacuous even if it is only loose by a factor of 2.

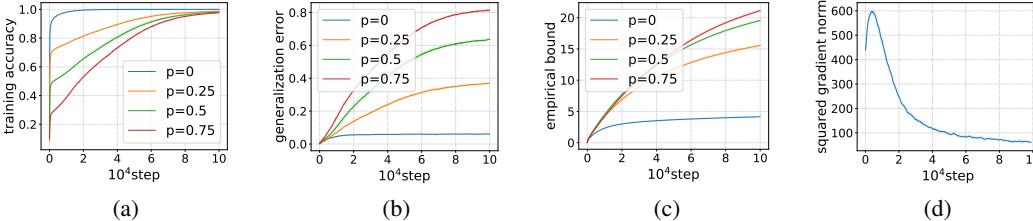

Figure 1: Training MLP with GLD ($\sigma_t = 0.2\sqrt{2}\gamma_t$) on a smaller version of MNIST with different random label portion $p$. (a) shows the training accuracy. (b) shows the generalization error, i.e., the gap between the 0/1 loss $\mathcal{L}^{01}$ on the training data and on the test data. (c) plots our bound in Theorem 9. (d) shows that for $p = 0$, the gradient norms become much smaller at later stages of training.

**Relax the step size constraint.** The condition on the step size in Theorem 11 may seem restrictive in the practical use.[5] We provide several ways to relax this constraint:

1. The proof of Theorem 11 still goes through if we replace $L$ with $\max_{i \in [n]} \|\nabla F(W_{t-1}, z_i)\|_2$ in the constraint.

2. The maximum gradient norm can be controlled by gradient clipping, i.e., multiplying $\frac{\min(C_L, \|\nabla F(W_{t-1}, z_i)\|_2)}{\|\nabla F(W_{t-1}, z_i)\|_2}$ to each $\nabla F(W_{t-1}, z_i)$.

3. Replacing the constant 20 with 2 in this constraint will only increase the constant of our bound from 8.12 to 84.4.

We also provide an experiment combining the above ideas to make our Theorem 11 applicable in the practical use (see Figure 4 in Appendix C).

## 4 GENERALIZATION OF CLD AND GLD WITH $\ell_2$ REGULARIZATION

In this section, we study the generalization error of Continuous Langevin Dynamics (CLD) with $\ell_2$ regularization. Throughout this section, we assume that the objective function over training set $S$ is defined as $F(w, S) = F_0(w, S) + \frac{\lambda}{2} \|w\|_2^2$, and moreover, the following assumption holds.

**Assumption 14.** *The loss function $\mathcal{L}$ and the original objective $F_0$ are $C$-bounded. Moreover, $F_0$ is differentiable and $L$-lipschitz.*

The Continuous Langevin Dynamics is defined by the following SDE:

$$\mathrm{d}W_t = -\nabla F(W_t, S)\,\mathrm{d}t + \sqrt{2\beta^{-1}}\,\mathrm{d}B_t, \quad W_0 \sim \mu_0, \tag{CLD}$$

where $(B_t)_{t\geq 0}$ is the standard Brownian motion on $\mathbb{R}^d$ and the initial distribution $\mu_0$ is the centered Gaussian distribution in $\mathbb{R}^d$ with covariance $\frac{1}{\lambda\beta}I_d$. We show that the generalization error of CLD is upper bounded by $O\left(e^{4\beta C} n^{-1} \sqrt{\beta/\lambda}\right)$, which is independent of the training time $T$ (Theorem 15). Furthermore, as $T$ goes to infinity, we have a tighter generalization error bound $O\left(\beta C^2 n^{-1}\right)$ (Theorem 39 in Appendix B). We also study the generalization of Gradient Langevin Dynamics (GLD), which is the discretization of CLD:

$$W_{k+1} = W_k - \eta \nabla F(W_k, S) + \sqrt{2\eta\beta^{-1}}\xi_k, \tag{GLD}$$

where $\xi_k$ is the standard Gaussian random vector in $\mathbb{R}^d$. By leveraging a result developed in Raginsky et al. (2017), we show that, as $K\eta^2$ tends to zero, GLD has the same generalization as CLD (see Theorems 15 and 39). We first formally state our first main result in this section.

---

[5]The condition $\gamma_t = O(\sigma_t/L)$ is also required in (Mou et al., 2018, Theorem 1)

**Theorem 15.** *Under Assumption 14, CLD (with initial probability measure $\mathrm{d}\mu_0 = \frac{1}{Z}e^{\frac{-\lambda\beta\|w\|^2}{2}}\,\mathrm{d}w$) has the following expected generalization error bound:*

$$\mathrm{err}_{gen} \leq \frac{2e^{4\beta C}CL}{n}\sqrt{\frac{\beta}{\lambda}\left(1 - \exp\left(-\frac{\lambda T}{e^{8\beta C}}\right)\right)}. \tag{7}$$

*In addition, if $\mathcal{L}$ is $M$-smooth and non-negative, by setting $\lambda\beta > 2$, $\lambda > 0$ and $\eta \in \left[0, 1 \wedge \frac{\lambda}{8M^2}\right)$, GLD (running $K$ iterations with the same $\mu_0$ as CLD) has the expected generalization error bound:*

$$\mathrm{err}_{gen} \leq 2C\sqrt{2KC_1\eta^2} + \frac{2CLe^{4\beta C}}{n}\sqrt{\frac{\beta}{\lambda}\left(1 - \exp\left(-\frac{\lambda\eta K}{e^{8\beta C}}\right)\right)}, \tag{8}$$

*where $C_1$ is a constant that only depends on $M$, $\lambda$, $\beta$, $b$, $L$ and $d$.*

The following lemma is crucial for establishing the above generalization bound for CLD. In particular, we need to establish a Log-Sobolev inequality for $\mu_t$, the parameter distribution at time $t$, for every time step $t > 0$. In contrast, most known LSIs only characterize the stationary distribution of the Markov process. The proof of the lemma can be found in Appendix B.

**Lemma 16.** *Under Assumption 14, let $\mu_t$ be the probability measure of $W_t$ in CLD (with $\mathrm{d}\mu_0 = \frac{1}{Z}e^{\frac{-\lambda\beta\|w\|^2}{2}}\,\mathrm{d}w$). Let $\nu$ be a probability measure that is absolutely continuous with respect to $\mu_t$. Suppose $\mathrm{d}\mu_t = \pi_t(w)\,\mathrm{d}w$ and $\mathrm{d}\nu = \gamma(w)\,\mathrm{d}w$. Then, it holds that*

$$\mathrm{KL}(\gamma, \pi_t) \leq \frac{\exp(8\beta C)}{2\lambda\beta}\int_{\mathbb{R}^d}\left\|\nabla\log\frac{\gamma(w)}{\pi_t(w)}\right\|_2^2\gamma(w)\,\mathrm{d}w.$$

We sketch the proof of Theorem 15, and the complete proof is relegated to Appendix B.

**Proof Sketch of Theorem 15** Suppose $S$ and $S'$ are two neighboring datasets. Let $(W_t)_{t\geq 0}$ and $(W_t')_{t\geq 0}$ be the process of CLD running on $S$ and $S'$, respectively. Let $\gamma_t$ and $\pi_t$ be the pdf of $W_t'$ and $W_t$. Let $F_S(w)$ denote $F(w, S)$. We have

$$\frac{\mathrm{d}}{\mathrm{d}t}\mathrm{KL}(\gamma_t, \pi_t) = \frac{-1}{\beta}\int_{\mathbb{R}^d}\gamma_t\left\|\nabla\log\frac{\gamma_t}{\pi_t}\right\|_2^2\,\mathrm{d}w + \int_{\mathbb{R}^d}\gamma_t\langle\nabla\log\frac{\gamma_t}{\pi_t}, \nabla F_S - \nabla F_{S'}\rangle\,\mathrm{d}w$$

$$\leq \frac{-1}{2\beta}\int_{\mathbb{R}^d}\gamma_t\left\|\nabla\log\frac{\gamma_t}{\pi_t}\right\|_2^2\,\mathrm{d}w + \frac{\beta}{2}\int_{\mathbb{R}^d}\gamma_t\|\nabla F_S - \nabla F_{S'}\|_2^2\,\mathrm{d}w.$$

$$\leq \frac{-\lambda}{e^{8\beta C}}\mathrm{KL}(\gamma_t, \pi_t) + \frac{2\beta L^2}{n^2} \qquad\qquad \text{(Lemma 16)}$$

Solving this inequality gives $\mathrm{KL}(\gamma_t, \pi_t) \leq \frac{1}{n^2\lambda}2\beta L^2 e^{8\beta C}(1 - e^{-\lambda t/e^{8\beta C}})$. Hence the generalization error of CLD can be bounded by $2C\sqrt{\frac{1}{2}\mathrm{KL}(\gamma_T, \pi_T)}$, which proves the first part. The second part of the theorem follows from Lemma 36 in Appendix B. ∎

Our second generalization bound for CLD (Theorem 39 in Appendix B) is

$$\mathrm{err}_{gen} \leq \frac{8\beta C^2}{n} + 4C\exp\left(\frac{-\lambda T}{e^{4\beta C}}\right)\sqrt{\beta C}.$$

The high level idea to prove this bound is very similar to that in Raginsky et al. (2017). We first observe that the (stationary) Gibbs distribution $\mu$ has a small generalization error. Then, we bound the distance from $\mu_t$ to $\mu$. In our setting, we can use the Holley-Strook perturbation lemma which allows us to bound the Logarithmic Sobolev constant, and we can thus bound the above distance easily.

## 5 FUTURE DIRECTIONS

In this paper, we prove new generalization bounds for a variety of noisy gradient-based methods. Our current techniques can only handle continuous noises for which we can bound the KL-divergence.

One future direction is to study the discrete noise introduced in SGD (in this case the KL-divergence may not be well defined). For either SGLD or CLD, if the noise level is small (i.e., $\beta$ is large), it may take a long time for the diffusion process to reach the stable distribution. Hence, another interesting future direction is to consider the local behavior and generalization of the diffusion process in finite time through the techniques developed in the studies of metastability (see e.g., Bovier et al. (2005); Bovier & den Hollander (2006); Tzen et al. (2018)). In particular, the technique may be helpful for further improving the bounds in Theorems 15 and 39 (when $T$ is not very large).

## 6 ACKNOWLEDGEMENT

We would like to thank Liwei Wang for several helpful discussions during various stages of the work. The research is supported in part by the National Natural Science Foundation of China Grant 61822203, 61772297, 61632016, 61761146003, and the Zhongguancun Haihua Institute for Frontier Information Technology and Turing AI Institute of Nanjing.

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

## A    PROOFS IN SECTION 3

### A.1    BAYES-STABILITY FRAMEWORK

**Lemma 17.** *Under Assumption 6, for any prior distribution $P$ not depending on the dataset $S = (z_1, \ldots, z_n)$, the generalization error is upper bounded by*

$$\left| \mathop{\mathbb{E}}_z \left[ \mathop{\mathbb{E}}_{w \sim P} \mathcal{L}(w, z) - \mathop{\mathbb{E}}_{w \sim Q_z} \mathcal{L}(w, z) \right] \right| + \left| \mathop{\mathbb{E}}_z \left[ \mathop{\mathbb{E}}_{w \sim P} \mathcal{L}(w) - \mathop{\mathbb{E}}_{w \sim Q_z} \mathcal{L}(w) \right] \right|,$$

*where $\mathcal{L}(w)$ denotes the population loss $\mathcal{L}(w, \mathcal{D})$.*

**Proof of Lemma 17** Let $\mathrm{err}_{\mathrm{train}} = \mathbb{E}_S \, \mathbb{E}_{w \sim Q_S} \mathcal{L}(w, S)$ and $\mathrm{err}_{\mathrm{test}} = \mathbb{E}_S \, \mathbb{E}_{w \sim Q_S} \mathcal{L}(w)$. We can rewrite generalization error as $\mathrm{err}_{\mathrm{gen}} = \mathrm{err}_{\mathrm{test}} - \mathrm{err}_{\mathrm{train}}$, where

$$\mathrm{err}_{\mathrm{test}} = \mathop{\mathbb{E}}_z \mathop{\mathbb{E}}_{w \sim Q_{(1,z)}} \mathcal{L}(w) = \mathop{\mathbb{E}}_z \mathop{\mathbb{E}}_{w \sim Q_z} \mathcal{L}(w) \qquad \text{(Assumption 6)}$$

$$= \mathop{\mathbb{E}}_z \int_{\mathbb{R}^d} (Q_z(w) - P(w)) \mathcal{L}(w) \, \mathrm{d}w + \int_{\mathbb{R}^d} P(w) \mathcal{L}(w) \, \mathrm{d}w.$$

and

$$\mathrm{err}_{\mathrm{train}} = \frac{1}{n} \sum_{i=1}^n \mathop{\mathbb{E}}_S \mathop{\mathbb{E}}_{w \sim Q_S} \mathcal{L}(w, z_i)$$

$$= \frac{1}{n} \sum_{i=1}^n \mathop{\mathbb{E}}_z \mathop{\mathbb{E}}_{w \sim Q_{(i,z)}} \mathcal{L}(w, z) = \mathop{\mathbb{E}}_z \mathop{\mathbb{E}}_{w \sim Q_z} \mathcal{L}(w, z) \qquad \text{(Assumption 6)}$$

$$= \mathop{\mathbb{E}}_z \int_{\mathbb{R}^d} (Q_z(w) - P(w)) \mathcal{L}(w, z) \, \mathrm{d}w + \int_{\mathbb{R}^d} P(w) \mathop{\mathbb{E}}_z \mathcal{L}(w, z) \, \mathrm{d}w \qquad \text{($P$ is a prior)}$$

$$= \mathop{\mathbb{E}}_z \int_{\mathbb{R}^d} (Q_z(w) - P(w)) \mathcal{L}(w, z) \, \mathrm{d}w + \int_{\mathbb{R}^d} P(w) \mathcal{L}(w) \, \mathrm{d}w. \qquad \text{(definition of $f(w)$)}$$

Thus, we have

$$|\mathrm{err}_{\mathrm{gen}}| = |\mathrm{err}_{\mathrm{test}} - \mathrm{err}_{\mathrm{train}}|$$

$$= \left| \mathop{\mathbb{E}}_z \int_{\mathbb{R}^d} (Q_z(w) - P(w)) \mathcal{L}(w) \, \mathrm{d}w - \mathop{\mathbb{E}}_z \int_{\mathbb{R}^d} (Q_z(w) - P(w)) \mathcal{L}(w, z) \, \mathrm{d}w \right|$$

$$\leq \left| \mathop{\mathbb{E}}_z \left[ \mathop{\mathbb{E}}_{w \sim Q_z} \mathcal{L}(w, z) - \mathop{\mathbb{E}}_{w \sim P} \mathcal{L}(w, z) \right] \right| + \left| \mathop{\mathbb{E}}_z \left[ \mathop{\mathbb{E}}_{w \sim Q_z} \mathcal{L}(w) - \mathop{\mathbb{E}}_{w \sim P} \mathcal{L}(w) \right] \right|.$$

∎

Now we are ready to prove Theorem 7, which we restate in the following.

**Theorem 7** (Bayes-Stability)**.** *Suppose the loss function $\mathcal{L}(w, z)$ is $C$-bounded and the learning algorithm is order-independent (Assumption 6), then for any prior distribution $P$ not depending on $S$, the generalization error is bounded by both $2C \, \mathbb{E}_z \left[ \sqrt{2\mathrm{KL}(P, Q_z)} \right]$ and $2C \, \mathbb{E}_z \left[ \sqrt{2\mathrm{KL}(Q_z, P)} \right]$.*

**Proof** By Lemma 17,

$$\mathrm{err}_{\mathrm{gen}} \leq \left| \mathop{\mathbb{E}}_z \left[ \mathop{\mathbb{E}}_{w \sim P} \mathcal{L}(w, z) - \mathop{\mathbb{E}}_{w \sim Q_z} \mathcal{L}(w, z) \right] \right| + \left| \mathop{\mathbb{E}}_z \left[ \mathop{\mathbb{E}}_{w \sim P} \mathcal{L}(w) - \mathop{\mathbb{E}}_{w \sim Q_z} \mathcal{L}(w) \right] \right|$$

$$\leq \mathop{\mathbb{E}}_z \left[ 2C \cdot \mathrm{TV}(P, Q_z) + 2C \cdot \mathrm{TV}(P, Q_z) \right] \qquad \text{($C$-boundedness)}$$

$$\leq 4C \mathop{\mathbb{E}}_z \left[ \sqrt{\frac{1}{2} \mathrm{KL}(P, Q_z)} \right] \qquad \text{(Pinsker's inequality)}$$

The other bound follows from a similar argument.                                                                    ∎

## A.2   Technical Lemmas

Now we turn to the proof of Theorem 11. The following lemma allows us to reduce the proof of algorithmic stability to the analysis of a single update step.

**Lemma 10.** *Let $(W_0, \ldots, W_T)$ and $(W'_0, \ldots, W'_T)$ be two independent sequences of random variables such that for each $t \in \{0, \ldots, T\}$, $W_t$ and $W'_t$ have the same support. Suppose $W_0$ and $W'_0$ follow the same distribution. Then,*

$$\mathrm{KL}(W_{\leq T}, W'_{\leq T}) = \sum_{t=1}^{T} \mathop{\mathbb{E}}_{w_{<t} \sim W_{<t}} [\mathrm{KL}(W_t | W_{<t} = w_{<t}, W'_t | W'_{<t} = w_{<t})],$$

*where $W_{\leq t}$ denotes $(W_0, \ldots, W_t)$ and $W_{<t}$ denotes $W_{\leq t-1}$.*

**Proof** By the chain rule of the KL-divergence,

$$\mathrm{KL}(W_{\leq t}, W'_{\leq t}) = \mathrm{KL}(W_{<t}, W'_{<t}) + \mathop{\mathbb{E}}_{w_{<t} \sim W_{<t}} [\mathrm{KL}(W_t | W_{<t} = w_{<t}, W'_t | W'_{<t} = w_{<t})].$$

The lemma follows from a summation over $t = 1, \ldots, T$. ∎

The following lemma (see e.g., (Duchi, 2007, Section 9)) gives a closed-form formula for the KL-divergence between two Gaussian distributions.

**Lemma 18.** *Suppose that $P = \mathcal{N}(\mu_1, \Sigma_1)$ and $Q = \mathcal{N}(\mu_2, \Sigma_2)$ are two Gaussian distributions on $\mathbb{R}^d$. Then,*

$$\mathrm{KL}(P, Q) = \frac{1}{2} \left( \mathrm{tr}(\Sigma_2^{-1} \Sigma_1) + (\mu_2 - \mu_1)^\top \Sigma_2^{-1} (\mu_2 - \mu_1) - d + \ln \frac{\det(\Sigma_2)}{\det(\Sigma_1)} \right).$$

The following lemma (Topsoe, 2000, Theorem 3) helps us to upper bound the KL-divergence.

**Definition 19.** *Let $P$ and $Q$ be two probability distributions on $\mathbb{R}^d$. The directional triangular discrimination from $P$ to $Q$ is defined as*

$$\Delta^*(P, Q) = \sum_{k=0}^{+\infty} 2^k \cdot \Delta\left(2^{-k} P + (1 - 2^{-k}) Q, Q\right),$$

*where*

$$\Delta(P, Q) = \int_{\mathbb{R}^d} \frac{(P(w) - Q(w))^2}{P(w) + Q(w)} \, \mathrm{d}w.$$

**Lemma 20.** *For any two probability distributions $P$ and $Q$ on $\mathbb{R}^d$,*

$$\mathrm{KL}(P, Q) \leq \ln 2 \cdot \Delta^*(P, Q).$$

Recall that $G$ is the set of all possible mini-batches. $G_n = \{B \in G : n \in B\}$ denotes the collection of mini-batches that contain $n$, while $\overline{G_n} = G \setminus G_n$. $\mathrm{diam}(A) = \sup_{x,y \in A} \|x - y\|$ denotes the diameter of set $A$. The following technical lemma upper bounds the KL-divergence between two Gaussian mixtures induced by sampling a mini-batch from neighbouring datasets.

**Lemma 21.** *Suppose that batch size $b \leq n/2$. $\{\mu_B : B \in G\}$ and $\{\mu'_B : B \in G\}$ are two collections of points in $\mathbb{R}^d$ labeled by mini-batches of size $b$ that satisfy the following conditions for some constant $\beta \in [0, \sigma]$:*

1. *$\|\mu_B - \mu'_B\| \leq \beta$ for $B \in G_n$ and $\mu_B = \mu'_B$ for $B \in \overline{G_n}$.*

2. *$\mathrm{diam}(\{\mu_B : B \in G\} \cup \{\mu'_B : B \in G\}) \leq \sigma/10$.*

*Let $p_{\mu,\sigma}$ denote the Gaussian distribution $\mathcal{N}(\mu, \frac{\sigma^2}{2} I_d)$. Let $P = \frac{1}{|G|} \sum_{B \in G} p_{\mu_B, \sigma}$ and $P' = \frac{1}{|G|} \sum_{B \in G} p_{\mu'_B, \sigma}$ be two mixture distributions over all mini-batches. Then,*

$$\mathrm{KL}(P, P') \leq \frac{8.23 b^2 \beta^2}{\sigma^2 n^2}.$$

**Proof of Lemma 21** By Lemma 20, $\mathrm{KL}(P, P')$ is bounded by

$$\ln 2 \cdot \Delta^* (P, P') = \ln 2 \cdot \sum_{k=0}^{+\infty} 2^k \cdot \Delta \left( 2^{-k} P + (1 - 2^{-k}) P', P' \right)$$

$$= \ln 2 \cdot \sum_{k=0}^{+\infty} 2^k \cdot \int_{\mathbb{R}^d} \frac{4^{-k} (P(w) - P'(w))^2}{2^{-k} P(w) + (2 - 2^{-k}) P'(w)} \, \mathrm{d}w.$$

The numerator of the above integrand is upper bounded by

$$4^{-k}(P - P')^2 = 4^{-k} \left( \frac{1}{|G|} \sum_{B \in G} (p_{\mu_B, \sigma} - p_{\mu'_B, \sigma}) \right)^2$$

$$= \frac{4^{-k} |G_n|^2}{|G|^2} \left( \frac{1}{|G_n|} \sum_{B \in G_n} (p_{\mu_B, \sigma} - p_{\mu'_B, \sigma}) \right)^2 \tag{9}$$

$$\leq \frac{4^{-k} b^2}{n^2} \cdot \frac{1}{|G_n|} \sum_{B \in G_n} (p_{\mu_B, \sigma} - p_{\mu'_B, \sigma})^2,$$

while the denominator can be lower bounded as follows:

$$2^{-k} P + (2 - 2^{-k}) P' \geq \frac{2^{-k}}{|G|} \sum_{B \in \overline{G_n}} p_{\mu_B, \sigma} + \frac{2 - 2^{-k}}{|G|} \sum_{B \in \overline{G_n}} p_{\mu'_B, \sigma}$$

$$= \frac{2}{|G|} \sum_{B \in \overline{G_n}} p_{\mu_B, \sigma} \qquad (\mu_B = \mu'_B \text{ for } B \in \overline{G_n})$$

$$= \frac{1}{|\overline{G_n}|} \cdot \frac{2(n - b)}{n} \sum_{B \in \overline{G_n}} p_{\mu_B, \sigma}$$

$$\geq \frac{1}{|\overline{G_n}|} \sum_{B \in \overline{G_n}} p_{\mu_B, \sigma}, \qquad (b \leq n/2)$$

which implies, by the convexity of $1/x$, that

$$\frac{1}{2^{-k} P + (2 - 2^{-k}) P'} \leq \frac{1}{\frac{1}{|\overline{G_n}|} \sum_{B \in \overline{G_n}} p_{\mu_B, \sigma}} \leq \frac{1}{|\overline{G_n}|} \sum_{B \in \overline{G_n}} \frac{1}{p_{\mu_B, \sigma}}. \tag{10}$$

Inequalities (9) and (10) together imply

$$\Delta \left( 2^{-k} P + (1 - 2^{-k}) P', P' \right) \leq \frac{4^{-k} b^2}{n^2 |\overline{G_n}| |G_n|} \sum_{A \in \overline{G_n}} \sum_{B \in G_n} \int_{\mathbb{R}^d} \frac{(p_{\mu_B, \sigma}(w) - p_{\mu'_B, \sigma}(w))^2}{p_{\mu_A, \sigma}(w)} \, \mathrm{d}w. \tag{11}$$

Now we bound the right-hand side of (11) for fixed $A$ and $B$. By applying a translation and a rotation, we can assume without loss of generality that $\mu_A = 0$, and the last $d - 2$ coordinates of $\mu_B$ and $\mu'_B$ are all zero. Note that the integral is unchanged when we project the space to the two-dimensional subspace corresponding to the first two coordinates. Thus, it suffices to prove a bound for $d = 2$. We rewrite (11) as

$$\Delta(2^{-k} P + (1 - 2^{-k}) P', P') \leq \frac{4^{-k} b^2}{n^2 |\overline{G_n}| |G_n|} \sum_{A \in \overline{G_n}} \sum_{B \in G_n} \frac{1}{\pi \sigma^2} \int_{\mathbb{R}^2} \frac{\left( e^{-\left\| \frac{w - \mu_B}{\sigma} \right\|^2} - e^{-\left\| \frac{w - \mu'_B}{\sigma} \right\|^2} \right)^2}{e^{-\left\| \frac{w}{\sigma} \right\|^2}} \, \mathrm{d}w. \tag{12}$$

Let $I$ be the integral in the right-hand side of (12). Note that $\left\|\frac{\mu_B}{\sigma}\right\|, \left\|\frac{\mu_B'}{\sigma}\right\| \leq 0.1$ and $\left\|\frac{\mu_B - \mu_B'}{\sigma}\right\| \leq \frac{\beta}{\sigma}$. Let $\delta = \frac{\beta}{\sigma}$ and $r = \left\|\frac{w}{\sigma}\right\|$. Our goal is to bound $\max_{\delta \in [0,0.1]}(I\delta^{-2})$. Let $(x)^+ = \max(x, 0)$. Since

$$I \leq \sigma^2 \int_0^\infty \frac{\max_{y \in [(r-0.1)^+, r+0.1]}(e^{-y^2} - e^{-(y+\delta)^2})^2}{e^{-r^2}} 2\pi r \, \mathrm{d}r.$$

We have

$$\max_{\delta \in [0,0.1]} \frac{I}{\delta^2} \leq \sigma^2 \int_0^\infty e^{r^2} 2\pi r \max_{y \in [(r-0.1)^+, r+0.1]} \max_{\delta \in [0,0.1]} \frac{(e^{-y^2} - e^{-(y+\delta)^2})^2}{\delta^2} \, \mathrm{d}r.$$

Let $\phi(y, \delta) = (\frac{e^{-y^2} - e^{-(y+\delta)^2}}{\delta})^2$, we make two claims which we will prove later:

1. For all $y, \delta \geq 0$, $\phi(y, \delta) \leq \frac{2}{e}$.
2. For all $y \geq \frac{1}{\sqrt{2}}$, $\phi(y, \delta)$ is non-increasing in $\delta$.

The above claims imply that:

1. For any $r \in \left[0, \frac{1}{\sqrt{2}} + 0.1\right]$, $\max_{y \in [(r-0.1)^+, r+0.1], \delta \in [0,0.1]}[\phi(y, \delta)] \leq \frac{2}{e}$.
2. For any $r \in \left(\frac{1}{\sqrt{2}} + 0.1, +\infty\right)$, we have

$$\max_{y \in [(r-0.1)^+, r+0.1], \delta \in [0,0.1]} \phi(y, \delta) \leq \max_{y \in [(r-0.1)^+, r+0.1]} \lim_{\delta \to 0}[\phi(y, \delta)]$$

$$= \max_{y \in [(r-0.1)^+, r+0.1]} 4y^2 e^{-2y^2}$$

$$= 4(r - 0.1)^2 e^{-2(r-0.1)^2}.$$

The last step holds since $y \mapsto y^2 e^{-2y^2}$ is decreasing on $\left[\frac{1}{\sqrt{2}}, +\infty\right)$.

Thus we have

$$\max_{\delta \in [0,0.1]} \frac{I}{\delta^2} \leq \sigma^2 \int_0^{\frac{1}{\sqrt{2}}+0.1} e^{r^2} 2\pi r \cdot \frac{2}{e} \, \mathrm{d}r + \sigma^2 \int_{\frac{1}{\sqrt{2}}+0.1}^{+\infty} e^{r^2} 2\pi r \cdot 4(r-0.1)^2 e^{-2(r-0.1)^2} \, \mathrm{d}r$$

$$\leq 18.6487\sigma^2.$$

Plugging the above into (12) gives

$$\Delta(2^{-k}P + (1 - 2^{-k})P', P') \leq \frac{4^{-k}b^2}{n^2\sigma^2\pi}\delta^2 \max_{\delta \in [0,0.1]}(I/\delta^2) \leq \frac{4^{-k}b^2}{n^2\pi} \cdot 18.6487\delta^2.$$

We conclude that

$$\mathrm{KL}(P, P') \leq \ln 2 \sum_{k=0}^{+\infty} 2^k \cdot \Delta(2^{-k}P + (1 - 2^{-k})P', P')$$

$$\leq \frac{37.2974 b^2 \delta^2 \ln 2}{n^2\pi} \leq \frac{8.23 b^2 \beta^2}{n^2\sigma^2}.$$

Finally, we prove the two claims used above:

1. For all $y, \delta \geq 0$, let $h(x) = e^{-x^2}$, we have $e^{-y^2} - e^{-(y+\delta)^2} = \int_{y+\delta}^y h'(t) \, \mathrm{d}t$. Since $|h'(t)| = |e^{-t^2}(-2t)| = e^{-t^2}(2t)$. Let $\frac{\partial}{\partial t}|h'(t)| = 0$, we have $t = 1/\sqrt{2}$. Thus, $|h'(t)| \leq e^{1/2}\sqrt{2}$ and $e^{-y^2} - e^{-(y+\delta)^2} \leq \delta\sqrt{2/e}$.

2. Suppose $y \geq 1/\sqrt{2}$, we have

$$\frac{\partial}{\partial \delta}\left(\frac{e^{-y^2} - e^{-(y+\delta)^2}}{\delta}\right) = \frac{e^{-(y+\delta)^2}(2y\delta + 2\delta^2 + 1) - e^{-y^2}}{\delta^2}$$

$$= \frac{e^{-y^2}}{\delta^2}[e^{-2y\delta-\delta^2}(2y\delta + 2\delta^2 + 1) - 1].$$

Let $g(y, \delta) = e^{-2y\delta-\delta^2}(2y\delta + 2\delta^2 + 1) - 1$. Note that

(a) $\lim_{\delta \to 0} \frac{\partial}{\partial \delta} \left( \frac{e^{-y^2} - e^{-(y+\delta)^2}}{\delta} \right) = 0.$

(b) $\frac{\partial}{\partial \delta} g(1/\sqrt{2}, \delta) = -4e^{-\delta(\delta+\sqrt{2})} \delta^2 (\delta + \sqrt{2}) < 0$ and $g(1/\sqrt{2}, 0) = 0.$

It implies that $\frac{\partial}{\partial \delta} \left( \frac{e^{-(1/\sqrt{2})^2} - e^{-(1/\sqrt{2}+\delta)^2}}{\delta} \right) \leq 0$ for $\delta > 0$. Since $\frac{\partial}{\partial y} g(y, \delta) = -4\delta^2 e^{-\delta(\delta+2y)} (\delta + y) < 0$ for $y \geq 0$, we conclude that for any $y \geq 1/\sqrt{2}$:

$$\frac{\partial}{\partial \delta} \left( \frac{e^{-y^2} - e^{-(y+\delta)^2}}{\delta} \right) \leq \frac{\partial}{\partial \delta} \left( \frac{e^{-(1/\sqrt{2})^2} - e^{-(1/\sqrt{2}+\delta)^2}}{\delta} \right) \leq 0.$$

$\blacksquare$

## A.3 MAIN THEOREM

Recall that SGLD on dataset $S$ is defined as

$$W_t \leftarrow W_{t-1} - \gamma_t \nabla_w F(W_{t-1}, S_{B_t}) + \frac{\sigma_t}{\sqrt{2}} \mathcal{N}(0, I_d).$$

Here $\gamma_t$ is the step size. $B_t = \{i_1, \ldots, i_b\}$ is a subset of $\{1, \ldots, n\}$ of size $b$, and $S_{B_t} = (z_{i_1}, \ldots, z_{i_b})$ is the mini-batch indexed by $B_t$. Recall that $F(w, S)$ denotes $\frac{1}{|S|} \sum_{i=1}^{|S|} F(w, z_i)$. We restate and prove Theorem 11 in the following.

**Theorem 11.** *Suppose that the loss function $\mathcal{L}$ is $C$-bounded and the objective function $F$ is $L$-lipschitz. Assume that the following conditions hold:*

1. *Batch size $b \leq n/2$.*

2. *Learning rate $\gamma_t \leq \sigma_t/(20L)$.*

*Then, the following expected generalization error bound holds for $T$ iterations of SGLD* (1):

$$\mathrm{err}_{gen} \leq \frac{8.12C}{n} \sqrt{\mathop{\mathbb{E}}_{S \sim \mathcal{D}^n} \left[ \sum_{t=1}^{T} \frac{\gamma_t^2}{\sigma_t^2} \mathbf{g}_e(t) \right]}, \qquad \text{(empirical norm)}$$

*where $\mathbf{g}_e(t) = \mathbb{E}_{w \sim W_{t-1}} [\frac{1}{n} \sum_{i=1}^{n} \|\nabla F(w, z_i)\|_2^2]$ is the empirical squared gradient norm, and $W_t$ is the parameter at step $t$ of SGLD.*

**Proof** By Theorem 7, we have

$$\mathrm{err}_{gen} \leq 2C \mathop{\mathbb{E}}_{z} \sqrt{2\mathrm{KL}(Q_z, P)} \qquad (13)$$

for any prior distribution $P$. In particular, we define the prior as $P(w) = \mathbb{E}_{\overline{S} \sim \mathcal{D}^{n-1}}[P_{\overline{S}}(w)]$, where $P_{\overline{S}}(w) = Q_{(\overline{S}, \mathbf{0})}$. By the convexity of KL-divergence,

$$\mathrm{KL}(Q_z, P) = \mathrm{KL}\left( \mathop{\mathbb{E}}_{\overline{S}}[Q_{(\overline{S}, z)}], \mathop{\mathbb{E}}_{\overline{S}}[Q_{(\overline{S}, \mathbf{0})}] \right) \leq \mathop{\mathbb{E}}_{\overline{S}} \left[ \mathrm{KL}\left( Q_{(\overline{S}, z)}, Q_{(\overline{S}, \mathbf{0})} \right) \right]. \qquad (14)$$

Fix a data point $z \in \mathcal{Z}$. Let $(W_t)_{t \geq 0}$ and $(W'_t)_{t \geq 0}$ be the training process of SGLD for $S = (\overline{S}, z)$ and $S' = (\overline{S}, \mathbf{0})$, respectively. Fix a time step $t$ and $w_{<t} = (w_0, \ldots, w_{t-1})$. Let $P_t$ and $P'_t$ denote the distribution of $W_t$ and $W'_t$ conditioned on $W_{<t} = w_{<t}$ and $W'_{<t} = w_{<t}$, respectively. By the definition of SGLD, we have $P_t = \frac{1}{|G|} \sum_{B \in G} p_{\mu_B}$ and $P'_t = \frac{1}{|G|} \sum_{B \in G} p_{\mu'_B}$, where $\mu_B = w_{t-1} - \gamma_t \nabla_w F(w_{t-1}, S_B)$, $\mu'_B = w_{t-1} - \gamma_t \nabla_w F(w_{t-1}, S'_B)$, and $p_\mu$ denotes the Gaussian distribution $\mathcal{N}(\mu, \frac{\sigma_t^2}{2} I_d)$. We note that:

1. $\|\mu'_B - \mu_B\| \leq \frac{\gamma_t \|\nabla F(w_{t-1}, z)\|_2}{b}$ for $B \in G_n$ and $\mu_B = \mu'_B$ for $B \in \overline{G_n}$.

2. $\mathrm{diam}(\{\mu'_B : B \in G\} \cup \{\mu_B : B \in G\}) \leq 2\gamma_t L \leq \sigma_t/10$.

By applying Lemma 21 with $\beta = \frac{\gamma_t \|\nabla F(w_{t-1}, z)\|_2}{b}$ and $\sigma = \sigma_t$,

$$\mathrm{KL}(P_t, P'_t) \leq \frac{8.23\gamma_t^2 \|\nabla F(w_{t-1}, z)\|_2^2}{\sigma_t^2 n^2}.$$

By Lemma 10,

$$\mathrm{KL}(W_{\leq T}, W'_{\leq T}) = \sum_{t=1}^{T} \mathop{\mathbb{E}}_{w_{<t} \sim W_{<t}} [\mathrm{KL}(P_t, P'_t)]$$

$$\leq \sum_{t=1}^{T} \mathop{\mathbb{E}}_{w \sim W_{t-1}} \left[ \frac{8.23\gamma_t^2 \|\nabla F(w, z)\|_2^2}{\sigma_t^2 n^2} \right],$$

which implies that

$$\mathrm{KL}(Q_S, Q_{S'}) = \mathrm{KL}(W_T, W'_T) \leq \mathrm{KL}(W_{\leq T}, W'_{\leq T})$$

$$\leq \frac{8.23}{n^2} \sum_{t=1}^{T} \frac{\gamma_t^2}{\sigma_t^2} \mathop{\mathbb{E}}_{w \sim W_{t-1}} \left[ \|\nabla F(w, z)\|_2^2 \right].$$

Together with (13) and (14), we have

$$\mathrm{err_{gen}} \leq 2C \mathop{\mathbb{E}}_{z} \sqrt{2 \mathop{\mathbb{E}}_{\overline{S}} \left[ \frac{8.23}{n^2} \sum_{t=1}^{T} \frac{\gamma_t^2}{\sigma_t^2} \mathop{\mathbb{E}}_{w \sim W_{t-1}} \left[ \|\nabla F(w, z)\|_2^2 \right] \right]}$$

$$\leq 2C \sqrt{2 \mathop{\mathbb{E}}_{S} \left[ \frac{8.23}{n^2} \sum_{t=1}^{T} \frac{\gamma_t^2}{\sigma_t^2} \mathop{\mathbb{E}}_{w \sim W_{t-1}} \left[ \|\nabla F(w, z_n)\|_2^2 \right] \right]}. \qquad \text{(concavity of } \sqrt{x})$$

Since SGLD is order-independent, we can replace $\nabla F(w, z_n)$ with $\nabla F(w, z_i)$ for any $i \in [n]$ in the right-hand side of the above bound. Our theorem then follows from the concavity of $\sqrt{x}$. Furthermore, if we bound $\mathrm{KL}(P, Q_z)$ instead of $\mathrm{KL}(Q_z, P)$ in the above proof, we obtain the following bound that depends on the *population squared gradient norm*:

$$\mathrm{err_{gen}} \leq \frac{8.12C}{n} \sqrt{\mathop{\mathbb{E}}_{\overline{S}} \left[ \sum_{t=1}^{T} \frac{\gamma_t^2}{\sigma_t^2} \mathop{\mathbb{E}}_{w \sim W'_{t-1}} \mathop{\mathbb{E}}_{z \sim D} \|\nabla F(w, z)\|_2^2 \right]}.$$

∎

## A.4 EXTENSION TO GENERAL NOISES

We can extend the generalization bounds in previous sections, which require the noise to be Gaussian, to other general noises, namely the family of log-lipschitz noises.

**Definition 22** (Log-Lipschitz Noises). *A probability distribution on $\mathbb{R}^d$ with density $p$ is $L$-log-lipschitz if and only if $\|\nabla \ln p(w)\| \leq L$ holds for any $w \in \mathbb{R}^d$. A random variable $\zeta$ is called an $L$-log-lipschitz noise if and only if it is drawn from an $L$-log-lipschitz distribution.*

The analog of SGLD, noisy momentum method (Definition 24), and noisy NAG (Definition 25) can be naturally defined by replacing the Gaussian noise $\zeta_t$ at each iteration with an independent $L$-log-lipschitz noise in the definition.

The following lemma is an analog of Lemma 21 under $L$-log-lipschitz noises. Recall that $G$ denotes a collection of mini-batches of size $b$. Lemma 23 readily implies the analogs of Theorems 11, 26 and 27 under more general noise distributions.

**Lemma 23.** *Suppose that batch size $b \leq n/2$ and $\mathcal{N}$ is an $L_{noise}$-log-lipschitz distribution on $\mathbb{R}^d$. $\{\mu_B : B \in G\}$ and $\{\mu'_B : B \in G\}$ are two collections of points in $\mathbb{R}^d$ that satisfy the following conditions for some constant $\beta \in \left[0, \frac{1}{L_{noise}}\right]$:*

*1. $\|\mu_B - \mu'_B\| \leq \beta$ for $B \in G_n$ and $\mu_B = \mu'_B$ for $B \in \overline{G_n}$.*

*2. $\mathrm{diam}(\{\mu_B : B \in G\} \cup \{\mu'_B : B \in G\}) \leq 1$.*

*For $\mu \in \mathbb{R}^d$, let $p_\mu$ denote the distribution of $\zeta + \mu$ when $\zeta$ is drawn from $\mathcal{N}$. Let $P = \frac{1}{|G|} \sum_{B \in G} p_{\mu_B}$ and $P' = \frac{1}{|G|} \sum_{B \in G} p_{\mu'_B}$ be mixture distributions over all mini-batches. Then,*

$$\mathrm{KL}(P, P') \leq \frac{C_0 b^2 \beta^2}{n^2}$$

*for some constant $C_0$ that only depends on $L_{noise}$.*

**Proof of Lemma 23** Following the same argument as in the proof of Lemma 21, we have

$$\mathrm{KL}(P, P') \leq \ln 2 \cdot \sum_{k=0}^{+\infty} 2^k \cdot \Delta(2^{-k}P + (1 - 2^{-k})P', P') \tag{15}$$

where

$$\Delta(2^{-k}P + (1 - 2^{-k})P', P') \leq \frac{4^{-k}b^2}{n^2|\overline{G_n}||G_n|} \sum_{A \in \overline{G_n}} \sum_{B \in G_n} \int_{\mathbb{R}^d} \frac{(p_{\mu_B}(w) - p_{\mu'_B}(w))^2}{p_{\mu_A}(w)} \, \mathrm{d}w. \tag{16}$$

Fixed $A \in \overline{G_n}$ and $B \in G_n$. Let $p_{\mathrm{noise}}$ denote the density of the noise distribution $\mathcal{N}$. Since $\|\mu_A - \mu_B\| \leq 1$ and $p_{\mathrm{noise}}$ is $L_{\mathrm{noise}}$-log-lipschitz, we have

$$p_{\mu_B}(w) = p_{\mathrm{noise}}(w - \mu_B) \leq p_{\mathrm{noise}}(w - \mu_A) \cdot e^{L_{\mathrm{noise}}\|\mu_A - \mu_B\|} \leq e^{L_{\mathrm{noise}}} p_{\mu_A}(w).$$

Similarly, since $\|\mu_B - \mu'_B\| \leq \beta$, we have

$$e^{-\beta L_{\mathrm{noise}}} p_{\mu_B}(w) \leq p_{\mu'_B}(w) \leq e^{\beta L_{\mathrm{noise}}} p_{\mu_B}(w).$$

Then, it follows from $\beta L_{\mathrm{noise}} \leq 1$ that

$$(p_{\mu_B}(w) - p_{\mu'_B}(w))^2 \leq (e^{\beta L_{\mathrm{noise}}} - 1)^2 p_{\mu_B}(w)^2 \leq \beta^2 L_{\mathrm{noise}}^2 p_{\mu_B}(w)^2.$$

Therefore, the integral on the righthand side of (16) can be upper bounded as follows:

$$\begin{aligned}
\int_{\mathbb{R}^d} \frac{(p_{\mu_B}(w) - p_{\mu'_B}(w))^2}{p_{\mu_A}(w)} \, \mathrm{d}w &\leq \beta^2 L_{\mathrm{noise}}^2 \int_{\mathbb{R}^d} \frac{p_{\mu_B}(w)^2}{p_{\mu_A}(w)} \, \mathrm{d}w \\
&\leq \beta^2 L_{\mathrm{noise}}^2 \int_{\mathbb{R}^d} p_{\mu_B}(w) \cdot e^{L_{\mathrm{noise}}} \, \mathrm{d}w \\
&= \beta^2 L_{\mathrm{noise}}^2 e^{L_{\mathrm{noise}}}.
\end{aligned}$$

Plugging the above inequality into (15) and (16) gives

$$\Delta(2^{-k}P + (1 - 2^{-k})P', P') \leq \frac{4^{-k}b^2}{n^2|\overline{G_n}||G_n|} \sum_{A \in \overline{G_n}} \sum_{B \in G_n} \beta^2 L_{\mathrm{noise}}^2 e^{L_{\mathrm{noise}}} = L_{\mathrm{noise}}^2 e^{L_{\mathrm{noise}}} \cdot \frac{4^{-k}b^2\beta^2}{n^2}.$$

and

$$\mathrm{KL}(P, P') \leq \ln 2 \cdot \sum_{k=0}^{+\infty} 2^k L_{\mathrm{noise}}^2 e^{L_{\mathrm{noise}}} \cdot \frac{4^{-k}b^2\beta^2}{n^2} = 2 \ln 2 L_{\mathrm{noise}}^2 e^{L_{\mathrm{noise}}} \cdot \frac{b^2\beta^2}{n^2}.$$

∎

A.5 EXTENSION TO OTHER GRADIENT-BASED METHODS

A.5.1 STABILITY BOUND FOR MOMENTUM AND NESTEROVS ACCELERATED GRADIENT

We adopt the formulation of Classical Momentum and Nesterov's Accelerated Gradient (NAG) methods in Sutskever et al. (2013) and consider the noisy versions of them.

**Definition 24** (Noisy Momentum Method). *Noisy Momentum Method on objective function $F(w, z)$ and dataset $S$ is defined as*

$$\begin{cases} V_t \leftarrow \eta V_{t-1} - \gamma_t \nabla_w F(W_{t-1}, S_{B_t}) + \zeta_t \\ W_t \leftarrow W_{t-1} + V_t \end{cases}$$

**Definition 25** (Noisy Nesterovs Accelerated Gradient). *Noisy Nesterovs Accelerated Gradient (NAG) on objective function $F(w, z)$ and dataset $S$ is defined as*

$$\begin{cases} V_t \leftarrow \eta V_{t-1} - \gamma_t \nabla_w F(W_{t-1} + \eta V_{t-1}, S_{B_t}) + \zeta_t, \\ W_t \leftarrow W_{t-1} + V_t. \end{cases}$$

In both definitions, $\gamma_t$ is the step size, mini-batch $B_t$ is drawn uniformly from $G$, $\zeta_t$ is a Gaussian noise drawn from $\mathcal{N}(0, \frac{\sigma_t^2}{2} I_d)$, and $\eta \in [0, 1]$ is the momentum coefficient.

**Theorem 26.** *Under the same assumptions on the loss function, objective function, batch size and learning rate as in Theorem 11, the generalization bounds in Theorem 11 still hold for noisy momentum method and noisy NAG.*

**Proof of Theorem 26** For any time step $t$ and $w_{<t} = (w_0, w_1, ..., w_{t-1})$, let $P_t$ and $P_t'$ denote the distribution of $W_t$ and $W_t'$ conditioned on $W_{<t} = w_{<t}$ and $W_{<t}' = w_{<t}$, respectively. By definition, we have $P_t = \frac{1}{|G|} \sum_{B \in G} p_{\mu_B}$ and $P_t' = \frac{1}{|G|} \sum_{B \in G} p_{\mu_B'}$.

If $t = 1$, for both noisy momentum method and noisy NAG, we have

$$\mu_B = w_{t-1} - \gamma_t \nabla_w F(w_{t-1}, S_B),$$
$$\mu_B' = w_{t-1} - \gamma_t \nabla_w F(w_{t-1}, S_B').$$

For $t > 1$, if noisy momentum method is used, we have

$$\mu_B = w_{t-1} + \eta(w_{t-1} - w_{t-2}) - \gamma_t \nabla_w F(w_{t-1}, S_B),$$
$$\mu_B' = w_{t-1} + \eta(w_{t-1} - w_{t-2}) - \gamma_t \nabla_w F(w_{t-1}, S_B').$$

Similarly, the following holds under noisy NAG:

$$\mu_B = w_{t-1} + \eta(w_{t-1} - w_{t-2}) - \gamma_t \nabla_w F(w_{t-1} + \eta(w_{t-1} - w_{t-2}), S_B),$$
$$\mu_B' = w_{t-1} + \eta(w_{t-1} - w_{t-2}) - \gamma_t \nabla_w F(w_{t-1} + \eta(w_{t-1} - w_{t-2}), S_B').$$

In either case, it can be verified that the conditions of Lemma 21 hold for $\beta = \frac{2\gamma_t L}{b}$ and $\sigma = \sigma_t$. The rest of the proof is the same as the proof of Theorem 11. ∎

A.5.2 STABILITY BOUND FOR ENTROPY-SGD

In the Entropy-SGD algorithm due to Chaudhari et al. (2017), instead of directly optimizing the original objective $F(w)$, we minimize the *negative local entropy* defined as follows:

$$-E(w, \gamma) = -\log \int_{\mathbb{R}^d} \exp\left(-F(w') - \frac{\gamma}{2} \|w - w'\|_2^2\right) \mathrm{d}w \tag{17}$$

Intuitively, a wider local minimum has a lower loss (i.e., $-E(w, \gamma)$) than sharper local minima. See Chaudhari et al. (2017) for more details. The Entropy-SGD algorithm invokes standard SGD to minimize the negative local entropy. However, the gradient of negative local entropy

$$-\nabla_w E(w, \gamma) = \gamma\left(w - \mathbb{E}_{w' \sim P}[w']\right), \qquad P(w') \propto \exp(-F(w') - \frac{\gamma}{2} \|w - w'\|_2^2) \tag{18}$$

is hard to compute. Thus, the algorithm uses exponential averaging to estimate the gradient in the SGLD loop; see Algorithm 1 for more details.

We have the following generalization bound for Entropy-SGD.

---

**Algorithm 1:** Entropy-SGD

**Input:** Training set $S = (z_1, .., z_n)$ and loss function $g(w, z)$.

**Hyper-parameters:** Scope $\gamma$, SGD learning rate $\eta$, SGLD step size $\eta'$ and batch size $b$.

1 **for** $t = 1$ *to* $T$ **do**
2     //SGD iteration
3     $W_{t,0}, \mu_{t,0} \leftarrow W_{t-1,K+1}$;
4     **for** $k = 0$ *to* $K - 1$ **do**
5        //SGLD iteration
6        $B_{t,k} \leftarrow$ mini-batch with size $b$;
7        $W_{t,k+1} \leftarrow W_{t,k} - \eta' \nabla_w g(W_{t,k}, S_{B_{t,k}}) + \eta' \gamma (W_{t-1,K+1} - W_{t,k}) + \sqrt{\eta'} \varepsilon \mathcal{N}(0, \frac{1}{2} I_d)$;
8        $\mu_{t,k+1} \leftarrow (1 - \alpha)\mu_{t,k} + \alpha W_{t,k}$;
9     **end**
10     $W_{t,K+1} \leftarrow W_{t,K} - \eta\gamma(W_{t,K} - \mu_{t,K})$;
11 **end**
12 **return** $W_{T,K+1}$;

---

**Theorem 27.** *Suppose that the loss function $\mathcal{L}$ is $C$-bounded and the objective function $F$ is $L$-lipschitz. If batch size $b \leq n/2$ and $\sqrt{\eta'} \leq \varepsilon/(20L)$, the following expected generalization error bound holds for Entropy-SGD:*

$$\mathrm{err}_{gen} \leq \frac{8.12 C \sqrt{\eta'}}{\varepsilon n} \sqrt{\mathbb{E}_S \left[ \sum_{t=1}^{T} \sum_{k=0}^{K-1} \mathbf{g}_{\mathrm{e}}(t, k) \right]}, \qquad \text{(empirical norm)}$$

*where $\mathbf{g}_{\mathrm{e}}(t, k) = \mathbb{E}_{w \sim W_{t,k}}[\frac{1}{n} \sum_{i=1}^{n} \|\nabla F(w, z_i)\|_2^2]$ is the empirical squared gradient norm, and $W_{t,k}$ denotes the training process with respect to $S$.*

*Since $\mathbf{g}_{\mathrm{e}}(t, k)$ is at most $L^2$, it further implies the generalization error of Entropy-SGD is bounded by $O\left(\frac{C\sqrt{\eta'}L}{\varepsilon n}\sqrt{TK}\right)$.*

**Proof of Theorem 27** Define the history before time step $(t, k)$ as follows:

$$W_{\leq(t,k)} = (W_{0,0}, ..., W_{0,K+1}, ..., W_{t-1,0}, ..., W_{t-1,K+1}, W_{t,0}, ..., W_{t,k}). \qquad (19)$$

Since $\mu$ is only determined by $W$, we only need to focus on $W$. This proof is similar to the proof of Theorem 11. By setting $P = \mathbb{E}_{\overline{S}}[Q_{(\overline{S}, \mathbf{0})}]$. Suppose $S = (\overline{S}, z)$ and $S' = (\overline{S}, \mathbf{0})$ are fixed, let $W$ and $W'$ denote their training process, respectively. Considering the following 3 cases:

1. $W_{t,0} \leftarrow W_{t-1,K+1}$: In this case, for a fixed $w_{\leq(t-1,K+1)}$, we have

$$\mathrm{KL}\left(W_{t,0}|W_{\leq(t-1,K+1)} = w_{\leq(t-1,K+1)}, W'_{t,0}|W'_{\leq(t-1,K+1)} = w_{\leq(t-1,K+1)}\right) = 0.$$

2. $W_{t,k+1} \leftarrow W_{t,k} - \eta' \nabla_w g(W_{t,k}, S_{B_{t,k}}) + \eta' \gamma (W_{t-1,K+1} - W_{t,k}) + \sqrt{\eta'} \varepsilon \mathcal{N}(0, \frac{1}{2} I_d)$: In this case, fix a $w_{\leq(t,k)}$, applying Lemma 21 gives

$$\mathrm{KL}\left(W_{t,k+1}|W_{\leq(t,k)} = w_{\leq(t,k)}, W'_{t,k+1}|W'_{\leq(t,k)} = w_{\leq(t,k)}\right) \leq \frac{8.23\eta' \|\nabla F(w_{t,k}, z)\|_2^2}{\varepsilon^2 n^2}.$$

3. $W_{t,K+1} \leftarrow W_{t,K} - \eta\gamma(W_t, K - \mu_{t,K})$: In this case, for a fixed $w_{\leq(t,K)}$, we have

$$\mathrm{KL}\left(W_{t,K+1}|W_{\leq(t,K)} = w_{\leq(t,K)}, W'_{t,K+1}|W'_{\leq(t,K)} = w_{\leq(t,K)}\right) = 0.$$

By applying Lemma 10, we have

$$\mathrm{KL}(W_{T,K+1}, W'_{T,K+1}) \leq \frac{8.23\eta'}{\varepsilon^2 n^2} \sum_{t=1}^{T} \sum_{k=0}^{K-1} \mathbf{g}_{\mathrm{e}}(t, k),$$

and Where $\mathbf{g}_{\mathrm{e}}(t, k)$ is the empirical squared gradient norm of the $k$-th SGLD iteration in the $t$-th SGD iteration, respectively. The rest of the proof is the same as the proof of Theorem 11. ∎

# B    PROOFS IN SECTION 4

## B.1    MARKOV SEMIGROUP AND LOG-SOBOLEV INEQUALITY

The continuous version of the noisy gradient descent method is the Langevin dynamics, described by the following stochastic differential equation:

$$\mathrm{d}W_t = -\nabla F(W_t)\,\mathrm{d}t + \sqrt{2\beta^{-1}}\,\mathrm{d}B_t, \quad W_0 \sim \mu_0, \tag{20}$$

where $B_t$ is the standard Brownian motion. To analyze the above Langevin dynamics, we need some preliminary knowledge about Log-Sobolev inequalities.

Let $p_t(w, y)$ denote the probability density function (i.e., probability kernel) describing the distribution of $W_t$ starting from $w$. For a given SDE such as (20), we can define the associated diffusion semigroup $\mathbf{P}$:

**Definition 28** (Diffusion Semigroup). *(see e.g., (Bakry et al., 2013, p. 39)) Given a stochastic differential equation (SDE), the associated diffusion semigroup $\mathbf{P} = (P_t)_{t \geq 0}$ is a family of operators that satisfy for every $t \geq 0$, $P_t$ is a linear operator sending any real-valued bounded measurable function $f$ on $\mathbb{R}^d$ to*

$$P_t f(w) = \mathbb{E}[f(W_t)|W_0 = w] = \int_{\mathbb{R}^d} f(y)p_t(w, \mathrm{d}y).$$

The semigroup property $P_{t+s} = P_t \circ P_s$ holds for every $t, s \geq 0$. Another useful property of $P_t$ is that it maps a nonnegative function to a nonnegative function. The carré du champ operator $\Gamma$ of this diffusion semigroup (w.r.t (20)) is (Bakry et al., 2013, p. 42)

$$\Gamma(f, g) = \beta^{-1}\langle \nabla f, \nabla g\rangle.$$

We use the shorthand notation $\Gamma(f) = \Gamma(f, f) = \beta^{-1}\|\nabla f\|_2^2$, and define (with the convention that $0\log 0 = 0$)

$$\mathrm{Ent}_\mu(f) = \int_{\mathbb{R}^d} f \log f\,\mathrm{d}\mu - \int_{\mathbb{R}^d} f\,\mathrm{d}\mu \log\left(\int_{\mathbb{R}^d} f\,\mathrm{d}\mu\right).$$

**Definition 29** (Logarithmic Sobolev Inequality). *(see e.g., (Bakry et al., 2013, p. 237)) A probability measure $\mu$ is said to satisfy a logarithmic Sobolev inequality LS($\alpha$) (with respect to $\Gamma$), if for all functions $f : \mathbb{R}^d \to \mathbb{R}^+$ in the Dirichlet domain $\mathbb{D}(\mathcal{E})$,*

$$\mathrm{Ent}_\mu(f) \leq \frac{\alpha}{2}\int_{\mathbb{R}^d} \frac{\Gamma(f)}{f}\,\mathrm{d}\mu.$$

*$\mathbb{D}(\mathcal{E})$ is the set of functions $f \in \mathbb{L}^2(\mu)$ for which the quantity $\frac{1}{t}\int_{\mathbb{R}^d} f(f - P_t f)\,\mathrm{d}\mu$ has a finite (decreasing) limit as $t$ decreases to 0.*

A well-known Logarithmic Sobolev Inequality is the following result for Gaussian measures.

**Lemma 30** (Logarithmic Sobolev Inequality for Gaussian measure). *(Bakry et al., 2013, p. 258) Let $\mu$ be the centered Gaussian measure on $\mathbb{R}^d$ with covariance matrix $\sigma^2 I_d$. Then $\mu$ satisfies the following LSI:*

$$\mathrm{Ent}_\mu(f) \leq \frac{\sigma^2}{2}\int_{\mathbb{R}^d} \frac{\|\nabla f\|_2^2}{f}\,\mathrm{d}\mu$$

Lemma 30 states that the centered Gaussian measure with covariance matrix $\sigma^2 I_d$ satisfies LS($\beta\sigma^2$) (with respect to $\Gamma$), where $\Gamma = \beta^{-1}\langle\nabla f, \nabla g\rangle$ is the carré du champ operator of the diffusion semigroup defined above.

Before proving our results, we need some known results from Markov diffusion process. It is well known that the invariant measure (Bakry et al., 2013, p. 10) of the above CLD is the Gibbs measure $\mathrm{d}\mu = \frac{1}{Z_\mu}\exp(-\beta F(w))\,\mathrm{d}w$ (Menz et al., 2014, (1.3)). In other words, $\mu$ satisfies $\int_{\mathbb{R}^d} P_t f\mathrm{d}\mu = \int_{\mathbb{R}^d} f\mathrm{d}\mu$ for every bounded positive measurable function $f$, where $P_t$ is the Markov semigroup in Definition 28. The following lemma by Holley and Stroock Holley & Stroock (1987) (see also (Bakry et al., 2013, p. 240)) allows us to determine the Logarithmic Sobolev constant of the invariant measure $\mu$.

**Lemma 31** (Bounded perturbation). *Assume that the probability measure $\nu$ satisfies $LS(\alpha)$ (with respect to $\Gamma$). Let $\mu$ be a probability measure such that $1/b \leq \mathrm{d}\mu/\mathrm{d}\nu \leq b$ for some constant $b > 1$. Then $\mu$ satisfies $LS(b^2\alpha)$ (with respect to $\Gamma$).*

In fact, Lemma 31 is a simple consequence of the following variational formula in the special case that $\phi(x) = x \log x$, which we will also need in our proof:

**Lemma 32** (Variational formula). *(see .g., (Bakry et al., 2013, p. 240)) Let $\phi : I \to \mathbb{R}$ on some open interval $I \subset \mathbb{R}$ be convex of class $\mathcal{C}^2$. For every (bounded or suitably integrable) measurable function $f : \mathbb{R}^d \to \mathbb{R}$ with values in $I$,*

$$\int_{\mathbb{R}^d} \phi(f) \, \mathrm{d}\mu - \phi\left(\int_{\mathbb{R}^d} f \, \mathrm{d}\mu\right) = \inf_{r \in I} \int_{\mathbb{R}^d} [\phi(f) - \phi(r) - \phi'(r)(f - r)] \, \mathrm{d}\mu. \tag{21}$$

It is worth noting the integrand of the right-hand side is nonnegative due to the convexity of $\phi$.

### B.2 LOGARITHMIC SOBOLEV INEQUALITY FOR CLD

Recall that $F_S(w) = F(w, S) := F_0(w, S) + \lambda \|w\|_2^2/2$ is the sum of the empirical original objective $F_0(w, S)$ and $\ell_2$ regularization. Let $\mathrm{d}\mu = \frac{1}{Z_\mu} \exp(-\beta F_S(w)) \, \mathrm{d}w$ be the invariant (Gibbs) measure of CLD, and $\nu$ is the centered Gaussian measure $\mathrm{d}\nu = \frac{1}{Z_\nu} \exp(-\beta\lambda \|w\|_2^2/2) \, \mathrm{d}w$. Invoking Lemma 30 with $\sigma^2 = \frac{1}{\lambda\beta}$ shows that $\nu$ satisfies $LS(1/\lambda)$ (with respect to $\Gamma$). Consider the density $h(w) = \frac{\mathrm{d}\mu}{\mathrm{d}\nu} = \frac{Z_\nu}{Z_\mu} \exp(-\beta F_0(w, S))$. If the original objective function $F_0$ is $C$-bounded, we have $\exp(-2\beta C) \leq h(w) \leq \exp(2\beta C)$. By applying Lemma 31 with $b = \exp(2\beta C)$, we have the following lemma.

**Lemma 33.** *Under Assumption 14, let $\Gamma(f, g) = \beta^{-1}\langle\nabla f, \nabla g\rangle$ be the carré du champ operator of the diffusion semigroup associated to CLD, and $\mu$ be the invariant measure of the SDE. Then, $\mu$ satisfies $LS(e^{4\beta C}/\lambda)$ with respect to $\Gamma$.*

Let $\mu_t$ be the probability measure of $W_t$. By definition of $P_t$, for any real-valued bounded measurable function $f$ on $\mathbb{R}^d$ and any $s, t \geq 0$,

$$\mathbb{E}_{w \sim \mu_{t+s}} [f(w)] = \mathbb{E}_{w \sim \mu_s} [P_t f(w)]. \tag{22}$$

In particular, if the invariant measure $\mu = \mu_\infty$ exists, we have

$$\mathbb{E}_{w \sim \mu} [f(w)] = \mathbb{E}_{w \sim \mu_\infty} [P_t f(w)] = \mathbb{E}_{w \sim \mu_{t+\infty}} [f(w)] = \mathbb{E}_{w \sim \mu} [P_t f(w)]. \tag{23}$$

The following lemma is crucial for establishing the first generalization bound for CLD. In fact, we establish a Log-Sobolev inequality for $\mu_t$, the parameter distribution at time $t$, for any time $t > 0$. Note that our choice of the initial distribution $\mu_0$ is important for the proof. [6]

**Lemma 34.** *Under Assumption 14, let $\mu_t$ be the probability measure of $W_t$ in* (CLD) *with initial probability measure $\mathrm{d}\mu_0 = \frac{1}{Z} e^{\frac{-\lambda\beta\|w\|^2}{2}} \, \mathrm{d}w$. Let $\Gamma$ be the carré du champ operator of diffusion semigroup associated to* (CLD). *Then, for any $f : \mathbb{R}^d \to \mathbb{R}^+$ in $\mathbb{D}(\mathcal{E})$:*

$$\mathrm{Ent}_{\mu_t}(f) \leq \frac{e^{8\beta C}/\lambda}{2} \int_{\mathbb{R}^d} \frac{\Gamma(f)}{f} \, \mathrm{d}\mu_t$$

**Proof** Let $\mu$ be the invariant measure of CLD. By Lemma 33 and Definition 29,

$$\mathrm{Ent}_\mu(f) \leq \frac{e^{4\beta C}}{2\lambda\beta} \int_{\mathbb{R}^d} \frac{\|\nabla f\|_2^2}{f} \, \mathrm{d}\mu. \tag{24}$$

---

[6] For arbitrary initial distribution, it is impossible to prove similar inequality for any $t \geq 0$ (unless the loss is strongly convex).

By applying Lemma 32 with $\phi(x) = x \log x$, we rewrite the left-hand side as

$$\text{Ent}_\mu(f) := \int_{\mathbb{R}^d} f \log f \, \mathrm{d}\mu - \int_{\mathbb{R}^d} f \, \mathrm{d}\mu \log \left( \int_{\mathbb{R}^d} f \, \mathrm{d}\mu \right)$$

$$= \inf_{r \in I} \int_{\mathbb{R}^d} [\phi(f) - \phi(r) - \phi'(r)(f - r)] \, \mathrm{d}\mu$$

$$= \inf_{r \in I} \int_{\mathbb{R}^d} [P_t(\phi(f) - \phi(r) - \phi'(r)(f - r))] \, \mathrm{d}\mu.$$

where the last equation holds by the definition of invariant measure $\int P_t f \, \mathrm{d}\mu = \int f \, \mathrm{d}\mu$. Thus, we have

$$\inf_{r \in I} \int_{\mathbb{R}^d} [P_t(\phi(f) - \phi(r) - \phi'(r)(f - r))] \, \mathrm{d}\mu = \text{Ent}_\mu(f) \le \frac{e^{4\beta C}}{2\lambda\beta} \int_{\mathbb{R}^d} \frac{\|\nabla f\|_2^2}{f} \, \mathrm{d}\mu, \qquad (25)$$

Let $\mu_t$ be the probability measure of $W_t$. Lemma 32 and (22) together imply that

$$\text{Ent}_{\mu_t}(f) = \inf_{r \in I} \int_{\mathbb{R}^d} [\phi(f) - \phi(r) - \phi'(r)(f - r)] \, \mathrm{d}\mu_t$$

$$= \inf_{r \in I} \int_{\mathbb{R}^d} [P_t(\phi(f) - \phi(r) - \phi'(r)(f - r))] \, \mathrm{d}\mu_0 \qquad (26)$$

Since $P_t(\phi(f) - \phi(r) - \phi'(r)(f - r)) \ge 0$[7] and $\frac{\mathrm{d}\mu_0}{\mathrm{d}\mu} \le \exp(2\beta C)$, we have

$$\text{Ent}_{\mu_t}(f) = \inf_{r \in I} \int_{\mathbb{R}^d} [P_t(\phi(f) - \phi(r) - \phi'(r)(f - r))] \frac{\mathrm{d}\mu_0}{\mathrm{d}\mu} \, \mathrm{d}\mu$$

$$\le \exp(2\beta C)\text{Ent}_\mu(f) \le \frac{e^{6\beta C}}{2\lambda\beta} \int_{\mathbb{R}^d} \frac{\|\nabla f\|_2^2}{f} \, \mathrm{d}\mu. \qquad (27)$$

Since $\frac{\mathrm{d}\mu}{\mathrm{d}\mu_0} \le \exp(2\beta C)$ and $\mu$ is the invariant measure, we conclude that

$$\text{Ent}_{\mu_t}(f) \le \frac{e^{6\beta C}}{2\lambda\beta} \int_{\mathbb{R}^d} \frac{\|\nabla f\|_2^2}{f} \, \mathrm{d}\mu = \frac{e^{6\beta C}}{2\lambda\beta} \int_{\mathbb{R}^d} P_t \left( \frac{\|\nabla f\|_2^2}{f} \right) \mathrm{d}\mu$$

$$= \frac{e^{6\beta C}}{2\lambda\beta} \int_{\mathbb{R}^d} P_t \left( \frac{\|\nabla f\|_2^2}{f} \right) \frac{\mathrm{d}\mu}{\mathrm{d}\mu_0} \, \mathrm{d}\mu_0$$

$$\le \frac{e^{8\beta C}}{2\lambda\beta} \int_{\mathbb{R}^d} P_t \left( \frac{\|\nabla f\|_2^2}{f} \right) \mathrm{d}\mu_0 \qquad (28)$$

$$= \frac{e^{8\beta C}}{2\lambda} \int_{\mathbb{R}^d} \frac{\beta^{-1} \|\nabla f\|_2^2}{f} \, \mathrm{d}\mu_t = \frac{e^{8\beta C}/\lambda}{2} \int_{\mathbb{R}^d} \frac{\Gamma(f)}{f} \, \mathrm{d}\mu_t$$

$$\blacksquare$$

**Lemma 16.** *Under Assumption 14, let $\mu_t$ be the probability measure of $W_t$ in CLD (with $\mathrm{d}\mu_0 = \frac{1}{Z} e^{\frac{-\lambda\beta\|w\|^2}{2}} \, \mathrm{d}w$). Let $\nu$ be a probability measure that is absolutely continuous with respect to $\mu_t$. Suppose $\mathrm{d}\mu_t = \pi_t(w) \, \mathrm{d}w$ and $\mathrm{d}\nu = \gamma(w) \, \mathrm{d}w$. Then it holds that:*

$$\text{KL}(\gamma, \pi_t) \le \frac{\exp(8\beta C)}{2\lambda\beta} \int_{\mathbb{R}^d} \left\| \nabla \log \frac{\gamma(w)}{\pi_t(w)} \right\|_2^2 \gamma(w) \, \mathrm{d}w. \qquad (29)$$

**Proof** Let $f(w) = \gamma(w)/\pi_t(w)$, by Lemma 34 and $\int_{\mathbb{R}^d} f \, \mathrm{d}\mu_t = 1$, we have

$$\int_{\mathbb{R}^d} f \log f \, \mathrm{d}\mu_t \le \frac{e^{8\beta C}}{2\lambda\beta} \int_{\mathbb{R}^d} \frac{\|\nabla f\|_2^2}{f} \, \mathrm{d}\mu_t \qquad (30)$$

---

[7] This is because $\phi$ is convex and $P_t$ is a positive operator.

We can see that the left-hand side is equal to $\mathrm{KL}(\gamma, \pi_t)$ [8], and the right-hand side is equal to

$$\frac{e^{8\beta C}}{2\lambda\beta} \int_{\mathbb{R}^d} \frac{\left\|\nabla \frac{\gamma(w)}{\pi_t(w)}\right\|_2^2}{\gamma(w)/\pi_t(w)} \pi_t(w) \, \mathrm{d}w = \frac{e^{8\beta C}}{2\lambda\beta} \int_{\mathbb{R}^d} \left\|\nabla \log \frac{\gamma(w)}{\pi_t(w)}\right\|_2^2 \gamma(w) \, \mathrm{d}w.$$

This concludes the proof. ∎

## B.3 THE DISCRETIZATION LEMMA FROM RAGINSKY ET AL. (2017)

Let $h(w, z) = F_0(w, z) + \frac{\lambda\|w\|_2^2}{2}$. We can rewrite $F_S(w) = \frac{1}{n}\sum_{i=1}^n h(w, z_i)$. Define $\mu_{S,k}$ and $\nu_{S,t}$ as the probability measure of $W_k$ (in GLD) and $W_t$ (in CLD), respectively. Raginsky et al. (2017) provided a bound of $\mathrm{KL}(\mu_{S,k}, \nu_{S,\eta K})$ under Assumption 35. This bound enables us to derive a generalization error bound for the discrete GLD from the bound for the continuous CLD. We use the assumption from Raginsky et al. (2017). Their work considers the following SGLD:

$$W_{k+1} = W_k - \eta g_S(W_k) + \sqrt{2\eta\beta^{-1}}\xi_k.$$

Where $g_S(w_k)$ is a conditionally unbiased estimate of the gradient $\nabla F_S(w_k)$. In our GLD setting, $g_S(W_k)$ is equal to $\nabla F_S(W_k)$.

**Assumption 35.** *Let $F_S(w) = \frac{1}{n}\sum_{i=1}^n h(w, z_i) = F_0(w, S) + \frac{\lambda}{2}\|w\|_2^2$.*

1. *The function $h$ takes non-negative real values, and there exist constants $A, B \geq 0$, such that*

   $$|h(0, z)| \leq A \qquad and \qquad \|\nabla h(0, z)\|_2 \leq B \qquad \forall z \in \mathcal{Z}.$$

2. *For each $z \in \mathcal{Z}$, the function $h(\cdot, z)$ is $M$-smooth: for some $M > 0$,*

   $$\|\nabla h(w, z) - \nabla h(v, z)\|_2 \leq M \|w - v\|_2, \qquad \forall w, v \in \mathbb{R}^d.$$

3. *For each $z \in Z$, the function $h(\cdot, z)$ is $(m, b)$-dissipative: for some $m > 0$ and $b \geq 0$,*

   $$\langle w, \nabla h(w, z)\rangle \geq m \|w\|_2^2 - b, \qquad \forall w \in \mathbb{R}^d.$$

4. *There exists a constant $\delta \in [0, 1)$, such that, for each $S \in \mathcal{Z}^n$,*

   $$\mathbb{E}[\|g_S(w) - \nabla F_S(w)\|_2^2] \leq 2\delta\left(M^2\|w\|_2^2 + B^2\right), \qquad \forall w \in \mathbb{R}^d.$$

5. *The probability law $\mu_0$ of the initial hypothesis $W_0$ has a bounded and strictly positive density $p_0$ with respect to the Lebesgue measure on $\mathbb{R}^d$, and*

   $$\kappa_0 := \log \int_{\mathbb{R}^d} e^{\|w\|_2^2} p_0(w) \, \mathrm{d}w < \infty.$$

**Lemma 36.** *(Raginsky et al., 2017, Lemma 7) Suppose that Assumption 35 holds and set $\mu_{S,0} = \nu_{S,0} = \mu_0$. Then, for any $k \in \mathbb{N}$ and any $\eta \in (0, 1 \wedge \frac{m}{4M^2})$, the following inequality holds*

$$\mathrm{KL}(\mu_{S,k}, \nu_{S,\eta k}) \leq (C_0\beta\delta + C_1\eta)k\eta,$$

*where $C_0$ and $C_1$ are constants that only depend on $M$, $\kappa_0$, $m$, $b$, $\beta$, $B$ and $d$.*

## B.4 PROOFS FOR MAIN THEOREMS

**Theorem 15.** *Under Assumption 14, CLD (with initial probability measure $\mathrm{d}\mu_0 = \frac{1}{Z} e^{\frac{-\lambda\beta\|w\|^2}{2}} \, \mathrm{d}w$) has the following expected generalization error bound:*

$$\mathrm{err}_{gen} \leq \frac{2e^{4\beta C}CL}{n}\sqrt{\frac{\beta}{\lambda}\left(1 - \exp\left(-\frac{\lambda T}{e^{8\beta C}}\right)\right)} \tag{31}$$

---

[8]Indeed, $\int_{\mathbb{R}^d} f \log f \, \mathrm{d}\mu_t = \int_{\mathbb{R}^d} \frac{\gamma}{\pi_t} \log(\frac{\gamma}{\pi_t})\pi_t \mathrm{d}w = \mathrm{KL}(\gamma, \pi_t)$

*In addition, if $F_0$ is also $M$-smooth and non-negative, by setting $\lambda\beta > 2$, $\lambda > 0$ and $\eta \in [0, 1 \wedge \frac{\lambda}{8M^2})$, the GLD (running $K$ iterations with the same $\mu_0$ as CLD) has the expected generalization error bound:*

$$\text{err}_{gen} \leq 2C\sqrt{2KC_1\eta^2} + \frac{2CLe^{4\beta C}}{n}\sqrt{\frac{\beta}{\lambda}\left(1 - \exp\left(-\frac{\lambda\eta K}{e^{8\beta C}}\right)\right)}, \tag{32}$$

*where $C_1$ is a constant that only depends on $M$, $\lambda$, $\beta$, $b$, $L$ and $d$.*

**Proof of Theorem 15** We apply the uniform stability framework. Suppose $S$ and $S'$ are two neighboring datasets that differ on exactly one data point. Let $(W_t)_{t\geq0}$ and $(W'_t)_{t\geq0}$ be the process of CLD running on $S$ and $S'$, respectively. Let $\gamma_t$ and $\pi_t$ be the pdf of $W'_t$ and $W_t$. We have

$$\begin{aligned}
\frac{d}{dt}\text{KL}(\gamma_t, \pi_t) &= \frac{d}{dt}\int_{\mathbb{R}^d} \gamma_t \log\frac{\gamma_t}{\pi_t} \, dw \\
&= \int_{\mathbb{R}^d}\left(\frac{d\gamma_t}{dt}\log\frac{\gamma_t}{\pi_t} + \gamma_t \cdot \frac{\pi_t}{\gamma_t} \cdot \frac{\frac{d\gamma_t}{dt}\pi_t - \gamma_t\frac{d\pi_t}{dt}}{\pi_t^2}\right) \, dw \\
&= \int_{\mathbb{R}^d}\left(\frac{d\gamma_t}{dt}\log\frac{\gamma_t}{\pi_t}\right) \, dw - \int_{\mathbb{R}^d}\left(\frac{\gamma_t}{\pi_t}\frac{d\pi_t}{dt}\right) \, dw
\end{aligned} \tag{33}$$

According to Fokker-Planck equation (see Risken (1996)) for CLD, we know that

$$\frac{\partial\gamma_t}{\partial t} = \frac{1}{\beta}\Delta\gamma_t + \nabla\cdot(\gamma_t\nabla F_{S'}), \qquad \frac{\partial\pi_t}{\partial t} = \frac{1}{\beta}\Delta\pi_t + \nabla\cdot(\pi_t\nabla F_S).$$

It follows that

$$\begin{aligned}
I &:= \int_{\mathbb{R}^d}\left(\frac{d\gamma_t}{dt}\log\frac{\gamma_t}{\pi_t}\right) \, dw \\
&= \int_{\mathbb{R}^d}\left(\frac{1}{\beta}\Delta\gamma_t + \nabla\cdot(\gamma_t\nabla F_{S'})\right)\log\frac{\gamma_t}{\pi_t} \, dw \\
&= \frac{-1}{\beta}\int_{\mathbb{R}^d}\langle\nabla\log\frac{\gamma_t}{\pi_t}, \nabla\gamma_t\rangle \, dw - \int_{\mathbb{R}^d}\langle\nabla\log\frac{\gamma_t}{\pi_t}, \gamma_t\nabla F_{S'}\rangle \, dw, \qquad \text{(integration by parts)}
\end{aligned}$$

and

$$\begin{aligned}
J &:= \int_{\mathbb{R}^d}\left(\frac{\gamma_t}{\pi_t}\frac{d\pi_t}{dt}\right) \, dw \\
&= \int_{\mathbb{R}^d}\frac{\gamma_t}{\pi_t}\left(\frac{1}{\beta}\Delta\pi_t + \nabla\cdot(\pi_t\nabla F_S)\right) \, dw \\
&= \frac{-1}{\beta}\int_{\mathbb{R}^d}\langle\nabla\frac{\gamma_t}{\pi_t}, \nabla\pi_t\rangle \, dw - \int_{\mathbb{R}^d}\langle\nabla\frac{\gamma_t}{\pi_t}, \pi_t\nabla F_S\rangle \, dw. \qquad \text{(integration by parts)}
\end{aligned}$$

Together with (33), we have

$$\begin{aligned}
\frac{d}{dt}\text{KL}(\gamma_t, \pi_t) &= I - J \\
&= \frac{-1}{\beta}\int_{\mathbb{R}^d}\left(\langle\frac{\nabla\gamma_t}{\gamma_t} - \frac{\nabla\pi_t}{\pi_t}, \nabla\gamma_t\rangle - \langle\frac{\nabla\gamma_t}{\pi_t} - \frac{\gamma_t\nabla\pi_t}{\pi_t^2}, \nabla\pi_t\rangle\right) \, dw \\
&\quad - \int_{\mathbb{R}^d}\left(\langle\nabla\log\frac{\gamma_t}{\pi_t}, \gamma_t\nabla F_{S'}\rangle - \frac{\gamma_t}{\pi_t}\langle\nabla\log\frac{\gamma_t}{\pi_t}, \pi_t\nabla F_S\rangle\right) \, dw \\
&= \frac{-1}{\beta}\int_{\mathbb{R}^d}\gamma_t\left\|\nabla\log\frac{\gamma_t}{\pi_t}\right\|_2^2 \, dw + \int_{\mathbb{R}^d}\gamma_t\langle\nabla\log\frac{\gamma_t}{\pi_t}, \nabla F_S - \nabla F_{S'}\rangle \, dw \\
&\leq \frac{-1}{2\beta}\int_{\mathbb{R}^d}\gamma_t\left\|\nabla\log\frac{\gamma_t}{\pi_t}\right\|_2^2 \, dw + \frac{\beta}{2}\int_{\mathbb{R}^d}\gamma_t\|\nabla F_S - \nabla F_{S'}\|_2^2 \, dw.
\end{aligned}$$

The last step holds because $\langle \mathbf{a}/\sqrt{\beta}, \mathbf{b}\sqrt{\beta} \rangle \leq \frac{\|\mathbf{a}\|_2^2}{2\beta} + \frac{\beta\|\mathbf{b}\|_2^2}{2}$. Since $\|\nabla F_S - \nabla F_{S'}\|_2^2 \leq \frac{4L^2}{n^2}$, by Lemma 16, we have

$$\mathrm{KL}(\gamma_t, \pi_t) \leq \frac{e^{8\beta C}}{2\lambda\beta} \int_{\mathbb{R}^d} \gamma_t \left\| \nabla \log \frac{\gamma_t}{\pi_t} \right\|_2^2 \, dw,$$

which implies

$$\frac{-\lambda}{e^{8\beta C}} \mathrm{KL}(\gamma_t, \pi_t) \geq \frac{-1}{2\beta} \int_{\mathbb{R}^d} \gamma_t \left\| \nabla \log \frac{\gamma_t}{\pi_t} \right\|_2^2 \, dw.$$

Hence,

$$\frac{d}{dt} \mathrm{KL}(\gamma_t, \pi_t) \leq \frac{-\lambda}{e^{8\beta C}} \mathrm{KL}(\gamma_t, \pi_t) + \frac{2\beta L^2}{n^2}, \quad \text{with } \mathrm{KL}(\gamma_0, \pi_0) = 0. \tag{34}$$

Solving this differential inequality gives

$$\mathrm{KL}(\gamma_t, \pi_t) \leq \frac{2\beta L^2 e^{8\beta C}(1 - e^{-\lambda t/e^{8\beta C}})}{n^2 \lambda}. \tag{35}$$

By Pinsker's inequality, we can finally see that

$$\sup_z |\mathbb{E}_{\mathcal{A}}[\mathcal{L}(W_T', z) - \mathcal{L}(W_T, z)]| \leq 2C\sqrt{\frac{1}{2}\mathrm{KL}(\gamma_T, \pi_T)} \leq \frac{2e^{4\beta C}CL}{n}\sqrt{\frac{\beta\left(1 - e^{-\frac{\lambda T}{e^{8\beta C}}}\right)}{\lambda}}.$$

By Lemma 4, the generalization error of CLD is bounded by the right-hand side of the above inequality.

Now, we prove the second part of the theorem. Let $(W_k)_{k\geq 0}$ and $(W_k')_{k\geq 0}$ be the (discrete) GLD processes training on $S$ and $S'$, respectively. Then for any $z \in \mathcal{Z}$:

$$\begin{aligned}
&|\mathbb{E}[\mathcal{L}(W_K, z)] - \mathbb{E}[\mathcal{L}(W_K', z)]| \\
&\leq 2C \cdot \mathrm{TV}(\mu_{S,K}, \mu_{S',K}) && (C\text{-boundedness}) \\
&\leq 2C \cdot (\mathrm{TV}(\mu_{S,K}, \nu_{S,\eta K}) + \mathrm{TV}(\nu_{S,\eta K}, \nu_{S',\eta K}) + \mathrm{TV}(\mu_{S',K}, \nu_{S',\eta K})).
\end{aligned}$$

Since $\lambda\beta > 2$ and $\lambda > \frac{1}{2}$, Assumption 35 holds with $A = C$, $B = L$, $m = \frac{\lambda}{2}$, $b = \frac{L^2}{2\lambda}$, $\delta = 0$ and $\kappa_0 = \frac{d}{2}\log\left(1 + \frac{2}{\lambda\beta-2}\right)$. By applying Pinskers inequality and Lemma 36, we have

$$\mathrm{TV}(\mu_{S,K}, \nu_{S,\eta K}) \leq \sqrt{\frac{1}{2}\mathrm{KL}(\mu_{S,K}, \nu_{S,\eta K})} \leq \sqrt{\frac{1}{2}KC_1\eta^2} \tag{36}$$

and

$$\mathrm{TV}(\mu_{S',K}, \nu_{S',\eta K}) \leq \sqrt{\frac{1}{2}\mathrm{KL}(\mu_{S',K}, \nu_{S',\eta K})} \leq \sqrt{\frac{1}{2}KC_1\eta^2}. \tag{37}$$

From (35), we have

$$\mathrm{TV}(\nu_{S,\eta K}, \nu_{S',\eta K}) \leq \sqrt{\frac{1}{2}\mathrm{KL}(\nu_{S,\eta K}, \nu_{S',\eta K})} \leq \sqrt{\frac{\beta L^2 e^{8\beta C}\left(1 - e^{-\frac{\lambda\eta K}{e^{8\beta C}}}\right)}{n^2\lambda}} \tag{38}$$

Combining (36), (37) and (38), we have

$$|\mathbb{E}[\mathcal{L}(W_K, z)] - \mathbb{E}[\mathcal{L}(W_K', z)]| \leq 2C\sqrt{2KC_1\eta^2} + \frac{2CLe^{4\beta C}}{n}\sqrt{\frac{\beta\left(1 - e^{-\frac{\lambda\eta K}{e^{8\beta C}}}\right)}{\lambda}} := \epsilon_n.$$

By Definition 3, GLD is $\epsilon_n$-uniformly stable. Applying Lemma 4 gives the generalization bound of GLD. ∎

**Lemma 37** (Exponential decay in entropy). *(Bakry et al., 2013, Theorem 5.2.1) The logarithmic Sobolev inequality LS($\alpha$) for the probability measure $\mu$ is equivalent to saying that for every positive function $\rho$ in $\mathbb{L}^1(\mu)$ (with finite entropy),*

$$\mathrm{Ent}_\mu(P_t\rho) \leq e^{-2t/\alpha}\mathrm{Ent}_\mu(\rho)$$

*for every $t \geq 0$.*

The following Lemma shows that $P_t(\frac{\mathrm{d}\mu_0}{\mathrm{d}\mu}) = \mu_t$ in our diffusion process.

**Lemma 38.** *Let* $\mathbf{P}$ *denote the diffusion semigroup of CLD. Let* $\mu$ *denote the invariant measure of* $P$ *and let* $\mu_t$ *denote the probability measure of* $W_t$. *Then* $P_t(\frac{\mathrm{d}\mu_0}{\mathrm{d}\mu}) = \mu_t$.

**Proof** Let $\mathrm{d}\mu = \mu(x) \, \mathrm{d}x$ and $\mathrm{d}\mu_t = \mu_t(x) \, \mathrm{d}x$. As shown in (Pavliotis, 2014, page 118), our diffusion process (Smoluchowski dynamics) is reversible, which means $\mu(x)p_t(x,y) = \mu(y)p_t(y,x)$. Thus for any $g(x)$, we have

$$\mathbb{E}_{x \sim P_t(\frac{\mathrm{d}\mu_0}{\mathrm{d}\mu})} [g(x)] = \int g(x)\mu(x)(P_t(d\mu_0/d\mu))(x)dx$$

$$= \int g(x)\mu(x)dx \int \mu_0(y)p_t(x,y)/\mu(y)dy$$

$$= \int g(x)\mu(x)dx \int \mu_0(y)p_t(y,x)/\mu(x)dy$$

$$= \int g(x)\mu(x)\mu_t(x)/\mu(x)dx = \mathbb{E}_{x \sim \mu_t} [g(x)].$$

Since $g$ is arbitrary, $P_t(\frac{\mathrm{d}\mu_0}{\mathrm{d}\mu})$ and $\mu_t$ must be the same. ∎

**Theorem 39.** *Suppose that* $n > 8\beta C$. *Under Assumption 14, CLD (with initial distribution* $\mathrm{d}\mu_0 = \frac{1}{Z}e^{-\frac{\lambda\beta\|w\|^2}{2}} \, \mathrm{d}w$*) has the following expected generalization error bound:*

$$\mathrm{err}_{gen} \leq \frac{8\beta C^2}{n} + 4C \exp\left(\frac{-\lambda T}{e^{4\beta C}}\right)\sqrt{\beta C}.$$

*In addition, if* $F_0$ *is also* $M$*-smooth and non-negative, by setting* $\lambda\beta > 2$, $\lambda > \frac{1}{2}$ *and* $\eta \in [0, 1 \wedge \frac{2\lambda-1}{8M^2})$*, the GLD process (running* $K$ *iterations with the same* $\mu_0$ *as CLD) has the expected generalization error bound:*

$$\mathrm{err}_{gen} \leq 2C\sqrt{2KC_1\eta^2} + \frac{8\beta C^2}{n} + 4C \exp\left(\frac{-\lambda\eta K}{e^{4\beta C}}\right)\sqrt{\beta C},$$

*where* $C_1$ *is a constant that only depends on* $M$, $\lambda$, $\beta$, $b$, $L$ *and* $d$.

**Proof of Theorem 39** Suppose $S$ and $S'$ are two datasets that differ on exactly one data point. Let $(W_t)_{t\geq 0}$ and $(W'_t)_{t\geq 0}$ be their processes, respectively. Let $\mathrm{d}\mu_t = \pi_t(w) \, \mathrm{d}w$ and $\mathrm{d}\mu'_t = \pi'_t(w) \, \mathrm{d}w$ be the probability measure of $W_t$ and $W'_t$, respectively. The invariant measure of CLD for $S$ and $S'$ are denoted as $\mu$ and $\mu'$, respectively. Recall that

$$\mathrm{d}\mu = \frac{1}{Z_\mu}e^{-\beta F_S(w)} \, \mathrm{d}w, \qquad \mathrm{d}\mu' = \frac{1}{Z_{\mu'}}e^{-\beta F_{S'}(w)} \, \mathrm{d}w.$$

The total variation distance of $\mu$ and $\mu'$ is

$$\mathrm{TV}(\mu, \mu') = \frac{1}{2}\int_{\mathbb{R}^d} \left|1 - \frac{\mathrm{d}\mu'}{\mathrm{d}\mu}\right| \, \mathrm{d}\mu$$

$$= \frac{1}{2}\int_{\mathbb{R}^d} \left|1 - \frac{Z_\mu}{Z_{\mu'}}\exp(-\beta(F_{S'}(w) - F_S(w)))\right| \frac{1}{Z_\mu}e^{-\beta F_S(w)} \, \mathrm{d}w. \tag{39}$$

Since $\frac{Z_\mu}{Z_{\mu'}}\exp(-\beta(F_{S'}(w) - F_S(w))) \in \left[e^{-\frac{4\beta C}{n}}, e^{\frac{4\beta C}{n}}\right]$ and $\frac{4\beta C}{n} < 1/2$, we have

$$\mathrm{TV}(\mu, \mu') \leq \max\left\{\frac{1}{2}\left(1 - e^{-\frac{4\beta C}{n}}\right), \frac{1}{2}\left(e^{\frac{4\beta C}{n}} - 1\right)\right\} \leq \frac{4\beta C}{n}. \tag{40}$$

Since $\mu$ and $\mu'$ satisfy $LS(e^{4\beta C/\lambda})$ (Lemma 33), applying Lemma 37 with $\rho = \frac{\mathrm{d}\mu_0}{\mathrm{d}\mu}$ and $\rho' = \frac{\mathrm{d}\mu'_0}{\mathrm{d}\mu'}$ and Lemma 38 yields:

$$\mathrm{KL}(\mu_t, \mu) \leq \exp\left(\frac{-2\lambda t}{e^{4\beta C}}\right)\mathrm{KL}(\mu_0, \mu), \qquad \mathrm{KL}(\mu'_t, \mu') \leq \exp\left(\frac{-2\lambda t}{e^{4\beta C}}\right)\mathrm{KL}(\mu'_0, \mu'). \tag{41}$$

Since $\mathrm{KL}(\mu_0, \mu)$ and $\mathrm{KL}(\mu_0', \mu')$ are upper bounded by $2\beta C$, Pinsker's inequality implies that $\mathrm{TV}(\mu_t, \mu)$ and $\mathrm{TV}(\mu_t', \mu')$ are upper bounded by $\sqrt{\exp\left(\frac{-2\lambda t}{e^{4\beta C}}\right)\beta C}$. Combining with (40) and note that $\mathrm{TV}(\mu_t, \mu_t') \leq \mathrm{TV}(\mu_t, \mu) + \mathrm{TV}(\mu, \mu') + \mathrm{TV}(\mu_t', \mu')$, we have

$$\sup_z |\mathbb{E}_{\mathcal{A}}[\mathcal{L}(W_T, z) - \mathcal{L}(W_T', z)]| \leq 2C \cdot \mathrm{TV}(\mu_t, \mu_t') \leq 4C\sqrt{\exp\left(\frac{-2\lambda t}{e^{4\beta C}}\right)\beta C} + \frac{8\beta C^2}{n}.$$

By Lemma 4, the generalization error of CLD is bounded by the right-hand side.

The proof for GLD proceeds in the same way as the second part of the proof of Theorem 15. ∎

## C    EXPERIMENT DETAILS

We first present the general setup of our experiments:

**Dataset:**    We use MNIST (LeCun et al., 1998) and CIFAR10 (Krizhevsky & Hinton, 2009) in our experiments.

**Neural network:**    In our experiments, we test two different neural networks: a smaller version of AlexNet (Krizhevsky et al., 2012) and MLP. The structures of the networks are similar to what are used in Zhang et al. (2017a).

- Small AlexNet: $k$ is the kernel size, $d$ is the depth of a convolution layer, fc($m$) is the fully-connected layer that has $m$ neurons. The ReLU activation are used in the first 6 layers.

| 1 | 2 | 3 | 4 | 5 | 6 | 7 |
|---|---|---|---|---|---|---|
| conv(k:5,d:64) | pool(k:3) | conv(k:5,d:192) | pool(k:3) | fc(384) | fc(192) | fc(10) |

- MLP: The MLP used in our experiment has 3 hidden layers, each having width 512. We also use ReLU as the activation function in MLP.

**Objective function:**    For a data point $z = (x, y)$ in MNIST, the objective function is

$$F(W, z) = -\ln(\mathrm{softmax}(\mathrm{net}_W(x))[y]),$$

where $\mathrm{softmax}(a)[i] = \frac{e^{a[i]}}{\sum_{j=1}^{10} e^{a[j]}}$, and $\mathrm{net}_W(x)$ is the output of the neural network (10 dimensional vector). Note that the objective function $F$ is exactly the cross-entropy loss.

**0/1 loss**    : The 0-1 loss $\mathcal{L}^{01}$ is defined as:

$$\mathcal{L}^{01}(W, (x, y)) = \begin{cases} 1 & (\arg\max_i \mathrm{net}_W(x)[i]) \neq y, \\ 0 & \text{otherwise.} \end{cases} \tag{42}$$

**Random labels:**    Suppose the dataset contains $n$ datapoint, and the corruption portion is $p$. We randomly select $n \cdot p$ data points, and replace their labels with random labels, as in Zhang et al. (2017a).

### C.1    EXPERIMENTAL RESULTS FOR GLD

The result of this experiment (see Figure 1) is discussed in Section 3.1. Here we present our implementation details.
We repeat our experiment 5 times. At every individual run, we first randomly sample 10000 data points from the complete MNIST training data. The initial learning rate $\gamma_0 = 0.003$. It decays 0.995 after every 60 steps, and it stops decaying when it is lower than 0.0005. During the training, we keep $\sigma_t = 0.2\sqrt{2}\gamma_t$. Recall that the empirical squared gradient norm $\mathbf{g}_e(t)$ in our bound (Theorem 9)

is $\mathbb{E}_{W_{t-1}}[\frac{1}{n} \sum_{i=1}^{n} \|\nabla f(W_{t-1}, z)\|^2]$. Since it is time-consuming to compute the exact $\mathbf{g}_e(t)$, in our experiment, we use an unbiased estimation instead. At every step, we randomly sample a mini-batch $B$ with batch size 200 from the training data, and use $\frac{1}{200} \sum_{i \in B} \|\nabla f(W_{t-1}, z_i)\|^2$ as $\mathbf{g}_e(t)$ to compute our bound in Figure 1. The estimation of $\mathbf{g}_e(t)$ at every step $t$ is shown in Figure 1(d). Since $\mathbf{g}_e(t)$ is not very stable, in our figure, we plot its moving average over a window of size 100 to make the curve smoother (i.e., $\mathbf{g}_{avg}(t) = \frac{1}{100} \sum_{\tau=t}^{t+100} \mathbf{g}_e(\tau)$).

## C.2 EXPERIMENTAL RESULTS FOR SGLD

In this subsection, we present some experiment results for running SGLD on both MNIST and CIFAR10 datasets, to demonstrate that our bound (see Theorem 11), in particular the sum of the empirical squared gradient norms along the training path, can distinguish normal dataset from dataset that contains random labels. As shown in Figure 3, the curves of our bounds look quite similar to the generalization curves. Due to the sub-optimal constants in our bound, the bound is currently greater than 1, and hence we omit the numbers on the y-axis.

We note that in our experiments presented in Figure 3, the learning rate that we choose is larger than that required by the second condition of Theorem 11. This is because the global Lipschitz constant $L$ is hard to estimate and the model is not able to fit training data under a very large noise. As discussed in Section 3.1, we can relax $\sigma_t \geq (20L)\gamma_t$ to $\sigma_t \geq (2 \max_{i \in [n]} \|\nabla F(W_{t-1}, z_i)\|)\gamma_t$. By applying gradient clipping trick, we can further relax this condition to

$$\sigma_t \geq \min\{C_L, (2 \max_{i \in [n]} \|\nabla F(W_{t-1}, z_i)\|)\}\gamma_t, \tag{43}$$

where $C_L$ is defined in Section 3.1. In order to show that our observation ("random > normal") still holds when the step size satisfies the requirement of our theory, we run an experiment that using gradient clipping trick with $C_L = 1$. The model is trained on a small subset of MNIST as fitting the original data set with random labels under such a large Gaussian noise is extremely slow. As shown in Figure 4, the experimental results remain unchanged when all the conditions of our bound are met.

These experiments indicate that the sum of squared empirical gradient norms is highly related to the generalization performance, and we believe by further optimizing the constants in our bound, it is possible to achieve a generalization bound that is much closer to the real generalization error.

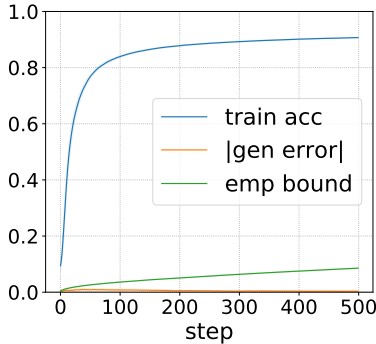

Figure 2: Training MLP with GLD ($\sigma_t = 0.2\gamma_t$) on the full MNIST dataset without label corruption. Learning rate $\gamma_t = 0.01 \cdot 0.95^{\lfloor t/60 \rfloor}$. Note that in the early stage, the testing accuracy is even higher than the training accuracy, thus we plot the absolute value of generalization error. As shown in this figure, even when the training accuracy approaches 90%, our bound is still relatively small.

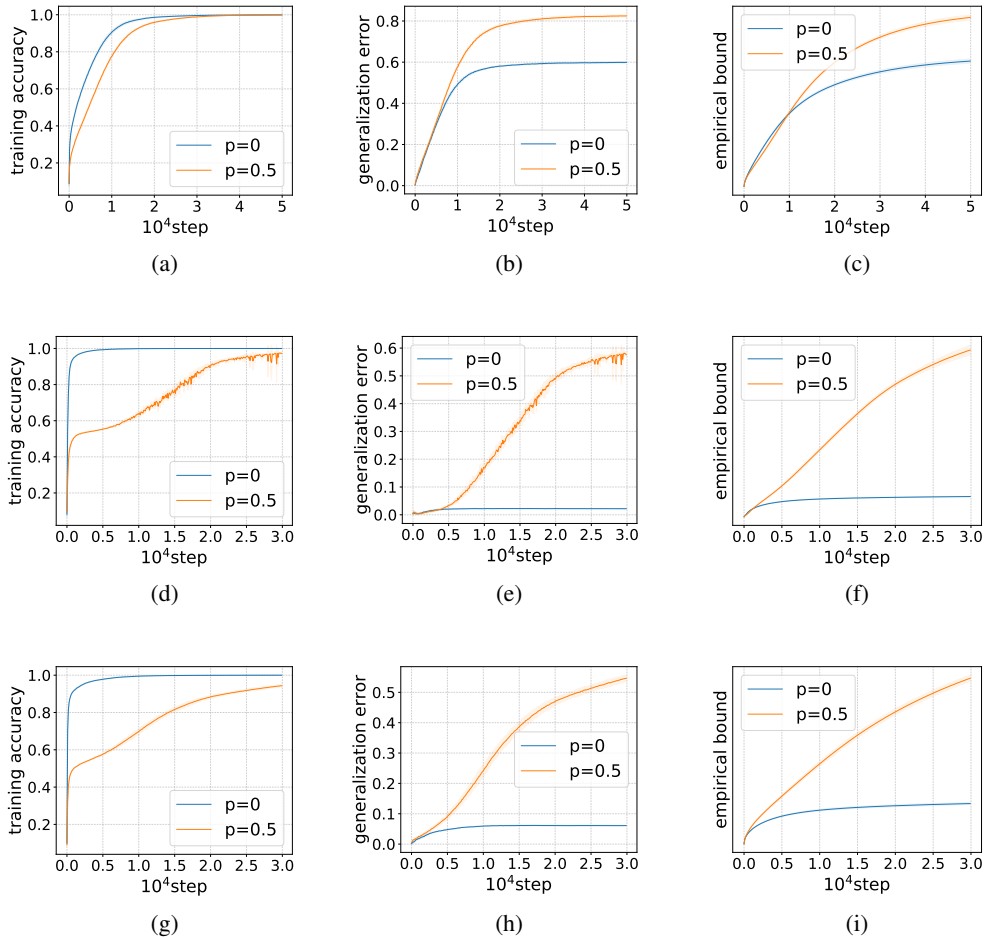

Figure 3: SGLD fitting random labels. The meaning of these plots are the same as those in Figure 1. (a-c): CIFAR10 + MLP; (d-f): MNIST+AlexNet; (g-i): MNIST+MLP; For each data set, only 5000 data points that randomly sampled from the complete dataset are used for training. Mini-batch size $b = 500$. Learning rate $\gamma_t = \max(0.0005, 0.003 \cdot 0.995^{\lfloor t/60 \rfloor})$. Noise level $\sigma_t = 0.002\sqrt{2}\gamma_t$.

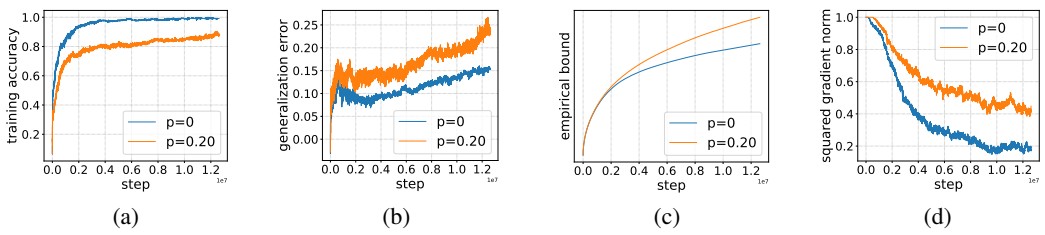

Figure 4: Training SGLD with AlexNet on a subset of MNIST (250 data points) with different random label portion $p$. The meanings of the y-labels are the same as that in Figure 1. We use the gradient clipping trick to force the gradient norms of every data points are within $C_L = 1$. We set $\sigma_t$ according to (43) with replacing "$\geq$" with "=". Mini-batch size $b = 10$. Learning rate $\gamma_t = \max(10^{-5}, 0.0005 \cdot 0.9^{\lfloor t/1000 \rfloor})$.

