# OpenReview forum: "On Generalization Error Bounds of Noisy Gradient Methods for Non-Convex Learning"
_ICLR.cc/2020/Conference — Accept (Poster)_

### Official Review · AnonReviewer2 · 2019-10-22
**Official Blind Review #2**

**Rating:** 6

**Review:**

This paper studies the generalization error bounds of stochastic gradient Langevin dynamics. The convexity of the loss function is not assumed. The author proposed "Bayes-stability" to derive generalization bound while taking the randomness of the algorithm into account. The generalization bound proposed in this paper applies to some existing problem setups. Also, the authors proposed the generalization bound of the continuous Langevin dynamics.

This is an interesting paper. Overall, the readability is high. The Bayes-stability is a significant contribution of this paper, and the theoretical analysis of the SGLD with non-Gaussian noise distribution will have a practical impact.

Some comments below:
- What is the function f of f(w,0)=0 above the equation (5)? Besides, the role of zero data point, i.e., f(w,0)=0, was not very clear.
- In the numerical results (b) and (c) of Figure 1, the scale in the y-axis was very different. What made the generalization bound so loose?
- In this paper, the developed theory was a general-purpose methodology. For deep neural networks, however, is there a meaningful insight obtained from the method developed in this paper?


**Experience Assessment:**

I have read many papers in this area.

**Review Assessment: Checking Correctness Of Derivations And Theory:**

I did not assess the derivations or theory.

**Review Assessment: Checking Correctness Of Experiments:**

I did not assess the experiments.

**Review Assessment: Thoroughness In Paper Reading:**

I made a quick assessment of this paper.

---

> ### Author Response · Authors · 2019-11-14
> **Response to Review #2 (Part 2)**
>
>
>
> 2."What made the generalization bound so loose?":
> Thanks for pointing this out. We list some possible reasons that may explain why our bound is larger than the real generalization error.
> a) Note that our bound (Theorem 9 and 11) hold for any trajectory-based output (i.e., the output could be any function of the training path $(W_{0},W_{1},...,W_{T})$) such as exponential moving average, average of the suffix of certain length or any other functions (see Remark 12). This is a stronger statement than an upper bound of the generalization error. In our experiment, we use $W_T$ (the last step parameter) as the output, while our Theorem 9 and 11 are upper bounds for the worst case trajectory-based output which may be larger.
> b) KL-divergence and  non-optimal constants:
> In our Theorem 7, we use the KL-divergence to bound the total variational distance (Pinsker's inequality). This step may not be very tight.
> Some of the constants such as $2\sqrt{2}$ in Theorem 9 may not be very tight.
> c) not large enough $\sigma_t$ in our experiments:
> Note the variance of Gaussian noise of SGLD (eq(1)) is very crucial to the actual bound. Our bound is much smaller if we use a not so small $\sigma_t$.
> However, we found in our experiment that if we choose a somewhat large variance, fitting the random lablelled training data can be extremely slow and we were not able to draw such a curve (the normal data can still be fit perfectly).
> Thus, we use a smaller noise level $\sigma_t \approx 0.3\eta_t$ instead.
> d) not large enough $n$:
> In fact, the data size we used is also very small ($n=10000$, it is not even a full mnist dataset) for the reason above (the convergence of training curve of the random labelled data is very slow when we use a somewhat larger variance, hence we choose a smaller subset of data).
> Here we provide an extra experiment on the full Mnist dataset ($n = 60000$) without label corruption:
>
> step | tra acc | gen_err | our bound
> 58    | 0.3000  | 0.0012   | 0.00845
> 116  | 0.6098  | 0.0014   | 0.01332
> 425  | 0.9006  | 0.0072   | 0.05081
>
> In this case, our bound is not vacuous.
> We also want to mention that it might be very difficult to obtain nonvacuous theorectical bound for randomly labelled data (no matter how large the $n$ is). For instance, consider a 10-classification task contains $100\%$ random labels. Since for any integer $n > 0$, one can find a deep neural network that can overfit the dataset. Thus, the training error is zero, and the generalization error is exactly the testing error, which must be larger than $90\%$. Therefore, any non-vacuous generalization bound should be in the range $[0.9,1]$. If the proven constant is, say, only 10 times larger, then the bound is already much larger than 1 and it can't be reduced by increasing $n$.
>
> Finally, we remark that our primary goal in this paper is not to make our bound non-vacuous numerically. Nevertheless, we believe by further optimizing some constants and chosing experimental setting more carefully, we can achieve a much tighter numerical bound, and this is left as an interesting furture work.
>
> 3."For deep neural networks,...": Thanks for the insightful question. Indeed, our bound is for general nonconvex learning and connects the generalization error and the sum of empirical gradient norms along the training path.
> Using this connection, we plan to investigate some concrete deep learning models, such as MLP or ResNet. In this case, we may be able to bound the gradient norm by some architecture-dependent factors.
> In fact, some recent papers study the landscape of the loss function and the training trajectory (e.g., [1][2][3][4]). It might be possible to use the insight of these results, combined with our gradient-norm-based generalization bound, to derive generalization bounds that depend on factors of the specific neural network such as the width, the depth or the least eigenvalue of certain Gram matrix ([4]). This is an interesting future direction.
>
> [1] Zhu et al., A convergence theory for deep learning via over-parameterization
> [2] Wu et al., Towards understanding generalization of deep learning: Perspective of loss landscapes
> [3] Tian., An analytical formula of population gradient for two-layered relu network and its applications in convergence and critical point analysis
> [4] Arora et al., Fine-Grained Analysis of Optimization and Generalization for Overparameterized Two-Layer Neural Networks

---

> > ### Comment · AnonReviewer2 · 2019-11-14
> > **Response**
> >
> > Thanks for the detailed comments. It will be good to add a brief summary of the second response about the upper bound, when this paper is published (in this ICLR or a future conference).

---

> > ### Comment · Area_Chair1 · 2019-11-14
> > **Question about bound**
> >
> > I have a question regarding the bound obtained for SGLD and stated in Theorem 11. The bound relies on two assumptions.  The first assumption does not seem to be very restrictive; however, the second assumption regarding the step size in terms of the Lipchitz constant is very restrictive.
> >
> > In this respect, I have two questions:
> > 1)	If one wants to relax the constraint on the step size, what will be the generalization bound for SGLD? It seems that one requires this strong assumption for the current proof.
> >
> > 2)	In Figure 2 on Page 30, you have plots that show the value of your bound along optimization trajectories for different models and data sets. In deep learning models that you have considered (like AlexNet), how do you estimate the Lipchitz constant so that you can make sure that the step size satisfies the conditions of Theorem 11? If I understand correctly, in fact, the step size used to produce these optimization trajectories DOES NOT satisfying the step size constraint. As such, the bound is not even applicable to these trajectories and the bound cannot be corrected simply by multiplying it by some constant. It seems also possible that, were you to have simulated trajectories under the very restrictive step-size constraint, the distinction between random and true labels may have looked different.
> >
> > Perhaps I'm missing something critical, but I'd like to understand this aspect.

---

> > > ### Author Response · Authors · 2019-11-15
> > > **Response to Area Chair #1**
> > >
> > > Thanks for the comments.
> > >
> > > We will answer your questions about the 2nd condition (small step size) stated in Theorem 11 as follows.
> > >
> > > (1) First, we remark the bound (Theorem 9) for GLD (a special case of SGLD) has NO constraint on the step size.
> > >
> > > (2) We need the step size constraint in the analysis of SGLD to bound the KL-divergence between two Gaussian mixtures for technical reasons.
> > > Since there is no closed-form formula of KL-divergence between Gaussian mixtures, we need Lemma 21 to provide an approximation for it. And the step size assumption ($\gamma_t \leq \sigma_t/(20L)$) in Theorem 11 is made for satisfying the 2nd condition of Lemma 21. We think it is possible to remove the constraint under certain other reasonable assumptions.
> > >
> > > (3) By the above discussion, we can in fact relax this assumption to $\gamma_t \leq \sigma_t/(20 \max_{i}\left\|\nabla F(W_{t-1},z_i)\right\|_2)$ and the 2nd condition of Lemma 21 still holds. Note that the training gradient norm $\max_{i}\left\|\nabla F(W_{t-1},z_i)\right\|_2$ is usually much smaller than the global Lipschitz constant $L$. Moreover, in the later stage of training, the gradient norm becomes very small (see Figure 1(d) in our paper), which enables a larger step size. Thus, if one relaxes the constraint to $\gamma_t \leq \sigma_t/(20 \max_{i}\left\|\nabla F(W_{t-1},z_i)\right\|_2)$, our bound still holds without change.
> > >
> > > (4) There are several other ways to further relax the step size constraint, by slightly adjusting the algorithm or analysis. For example, if the gradient clipping trick is used (i.e., multiplying $\frac{\min(C_L,\left\|\nabla F(W_{t-1},z_i)\right\|_2)}{\left\|\nabla F(W_{t-1},z_i)\right\|_2}$ to each $\nabla F(W_{t-1},z_i)$, where $C_L$ is not very large), the constraint can be further relaxed to $\gamma_t \leq \sigma_t/(20 C_{L})$ without affecting the bound. Replacing the constant $20$ with $2$ in this constraint will only increase the constant of our bound from $8.12$ to $84.4$.
> > >
> > > We don't actually set learning rate according to the above constraints precisely in our experiments. Hence, the experiment setting showed in Figure 2 does not match perfectly with the conditions in Theorem 11. In our submission, we have explicitly admitted this point in Appendix C.2. Nevertheless, we suspect that the conclusion of our lemmas should still hold (or hold for most steps) even without step size constraints, under certain other reasonable assumptions. It is an intriguing future direction.
> > >
> > > Finally, we want to mention that many works that study the trajectories of SGD or SGLD require some assumption on small step sizes (e.g., [1][2][3][4] and many works in the optimization literature), since most such analysis are inherently local. For example, in [1], they make the same assumption on the step size ($\gamma_t = O(\sigma_t / L)$) in their stability bound for SGLD (see Theorem 1 in [1]). Nevertheless, it would be an interesting and important future work to try to relax or remove these constraints that do not perfectly match the practice. We will discuss the step size constraint in more details in the next version.
> > >
> > > [1] Mou et al., Generalization bounds of SGLD for non-convex learning: Two theoretical viewpoints
> > > [2] Hardt et al., Train faster, generalize better: stability of stochastic gradient descent
> > > [3] Kuzborskij et al., Data-Dependent Stability of Stochastic Gradient Descent
> > > [4] Arora et al., Fine-Grained Analysis of Optimization and Generalization for Overparameterized Two-Layer Neural Networks

---

> ### Author Response · Authors · 2019-11-14
> **Response to Review #2 (Part 1)**
>
> Thanks for your careful review and insightful comments.
>
> 1."What is the function $f$ of $f(w,\mathbf{0})=0$": Sorry for the typo. $f$ should be capital $F$. The zero data point $\mathbf{0}$ is a synthetic data point (just a symbol). It is defined as a zero constant function (i.e., $F(w,\mathbf{0}):= 0$ for all $w\in\mathbb{R}^d$). Note that we don't need to care about what the zero data point $\mathbf{0}$ look like, because the only thing we use is its objective function's gradient $\nabla F(w,\mathbf{0})$ (which is also zero) in our analysis. The zero data point $\mathbf{0}$ is constructed for the convenience of defining the prior distribution $P$.

---

### Official Review · AnonReviewer3 · 2019-10-23
**Official Blind Review #3**

**Rating:** 6

**Review:**

In this paper, the authors provide new generalization analysis of (stochastic) gradient langevin dynamics in a nonconvex learning setting. The results are largely based on and improves the analysis in Mou et al. (2018). In more details,  Theorem 11 improves the corresponding generalization bound in Mou et al. (2018) by replacing the uniform Lipschitz constant by the expected empirical gradient norm, which can be smaller than the Lipschitz constant. The authors also argue this can distinguish normal data from randomly labelled data with experiments. The authors further studied the setting with an l_2 regularizer and derived improved result applicable to the case with infinite number of iterations, in which case the results in Mou et al. (2018) can diverge. These results are derived by a new bayes-stability method.

A drawback is that the results are only applicable to gradient methods in Section 4, i.e., using all examples in the gradient calculation. It would be interesting to see how the generalization bound would be for the stochastic counterparts.

The authors assume \lambda>1/2 in deriving (8). In practice, the regularization parameter should be set to be small enough to achieve a small test error. Therefore, eq (8) may not be quite interesting.

----------------------
After rebuttal:

I have read the authors' response. I would like to keep my original score.

**Experience Assessment:**

I have read many papers in this area.

**Review Assessment: Checking Correctness Of Derivations And Theory:**

I assessed the sensibility of the derivations and theory.

**Review Assessment: Checking Correctness Of Experiments:**

I did not assess the experiments.

**Review Assessment: Thoroughness In Paper Reading:**

I read the paper at least twice and used my best judgement in assessing the paper.

---

> ### Author Response · Authors · 2019-11-10
> **Response to Review #3**
>
> Thanks for your careful review and insightful comments.
> Indeed, the analysis can be extended to stochastic gradient (at the cost of an extra additive term) and the constant $1/2$ in the condition $\lambda>1/2$ can be relaxed to any small constant $c>0$ by slightly changing the proof (for ease of calculation and convenience, we chose the constant $1/2$ in the original submission). Now, we explain the details.
>
> 1.Extending Theorem 15 to stochastic gradient: The key step is to apply Lemma 36 in Appendix B.3. By Lemma 36, we have $KL(\mu_{S,K}, \nu_{S,\eta K}) <= (C_0 \beta \delta + C_1 \eta) K \eta$, where $\delta$ is the constant in the 4th condition of Assumption 35. In the full gradient case, the 4th condition of Assumption 35 holds with $\delta = 0$. In the stochastic gradient case, it holds with $\delta=1/(2 * \textrm{batch size})$. Hence, we need an extra additive $2C\sqrt{C_0K\eta / \textrm{batch size}}$ term in the generalization bound (eq(8)), when applying to stochastic gradient settings. Here a batch of data are i.i.d. drawn from full dataset $S$. Note that when batch size becomes larger, the extra term vanishes, and it matches the full gradient case bound (eq(8)).
>
> 2.Relax the condition $\lambda > 1/2$: Thanks for you to point it out! Here we only need to slightly modify our original proof of  Theorem 15. In particular, we show that $\lambda$ could be an arbitrary small positive number. Note that in the original proof of Theorem 15, $\lambda > 1/2$ is only used for satisfying the 2nd condition of Assumption 35 (hold with $m = (2\lambda-1)/(2)$, $b = L^2/2$, and m is required to be greater than 0 in this assumption, thus we set $\lambda > 1/2$ in our original proof). Note that the 2nd condition also holds with $m = \lambda /2$, $b = L^2/(2\lambda)$, where $\lambda$ could be any positive real number. Thus we only need to replace the statement "Assumption 35 holds with..." in page 26 with "Assumption 35 holds with $A = C, B = L, m = \lambda /2, b = L^2/(2\lambda)$" and change the upper bound of learning rate $\eta$ in Theorem 15 from $(2\lambda-1)/(8M^2)$ to $\lambda/(8M^2)$.

---

> > ### Comment · AnonReviewer3 · 2019-11-13
> > **Response**
> >
> > Thanks for your reply. Perhaps you mean the third assumption of Assumption 35. I now see this assumption also holds with $m=\lambda/2,b=L^2/(2\lambda)$ and the regularization parameter can be relaxed. I am also happy to see Theorem 15 also holds for stochastic gradient.

---

> > > ### Author Response · Authors · 2019-11-14
> > > **Re: Response**
> > >
> > > Thanks for your reply!  It is indeed the third assumption.  Sorry for this typo.

---

### Official Review · AnonReviewer1 · 2019-10-24
**Official Blind Review #1**

**Rating:** 6

**Review:**

This paper aims at developing a better understanding of generalization error for increasingly prevalent non-convex learning problems. For many such problems, the existing generalization bounds in the statistical learning theory literature are not very informative. To address these issues, the paper explores algorithm-specific generalization bounds,  especially focusing on various types of noisy gradient methods.

The paper employs a framework that combines uniform stability and PAC-Bayesian theory to obtain generalization bound for the noisy gradient methods. For gradient Langevin dynamic (GLD) and stochastic gradient Langevin dynamics (SGLD), using this Bayes-Stability framework, the paper obtains a generalization bound on the expected generalization error that scales with the expected empirical squared gradient norm. As argued in the paper, this provides an improvement over the existing bounds in the literature. Furthermore, this bound enables the treatment of the setting with noisy labels. For this setting the expected empirical squared gradient norm along the optimization path is higher, leading to worse generalization bound.

The paper then extends their results to the setting where an $\ell_2$ regularization is added to the non-convex objective. By using a new Log-Sobolev inequality for the parameter distribution at time t, the paper obtains new generalization bounds for continuous Langevin dynamic (CLD). These bounds subsequently provide bounds for GLD as well.

The paper demonstrates the utility of their generalization bound via empirical evaluation on MNIST and CIFAR dataset. The obtained generalization bounds are informative as they appear to capture the trend in the generalization error.

Overall, the paper is very well written with a clear comparison with the existing generalization bounds. The results in the paper are interesting and novel. That said, the discussion in the introduction and abstract appears a bit misleading as it gives the impression that this is the first paper that combines the ideas from stability and PAC-Bayesian theory to obtain generalization bounds. This is not the case, e.g. see [1].

As noted by the authors, some of the bounds obtained in this paper share similarities with one of the bounds in Mou et al.  as all these bounds contain the expected empirical squared gradient norm. The bound in Mou et al. holds with high probability and decays as $O(1/\sqrt{n})$, whereas the bounds in this paper are on expected generalization error and decay as $O(1/n)$. Could authors comment on extending their results to hold with high probability and how it would affect their bounds?

[1] Rivasplata et al., PAC-Bayes bounds for stable algorithms with instance-dependent priors.


----------------------- Post author response -------------

Thank you for addressing my comments. I have decided to keep my original score unchanged.

**Experience Assessment:**

I have read many papers in this area.

**Review Assessment: Checking Correctness Of Derivations And Theory:**

I carefully checked the derivations and theory.

**Review Assessment: Checking Correctness Of Experiments:**

I carefully checked the experiments.

**Review Assessment: Thoroughness In Paper Reading:**

I read the paper at least twice and used my best judgement in assessing the paper.

---

> ### Author Response · Authors · 2019-11-15
> **Response to Review #1**
>
> Thanks for your careful review and insightful comments!
>
> Regarding high-probability bounds, we note that our proof of Theorem 11 can be adapted to recover the previous bound of $O(LC \sqrt{T}/n)$ in (Mou et al., 2018, Theorem 1), with the expected squared gradient norm term relaxed to $L^2$, using the uniform stability framework. Then, applying the recent results of [Feldman and Vondrak, 2019, Theorem 1] gives a generalization error bound of $\tilde O(LC \sqrt{T}/n + 1/\sqrt{n})$ that holds with high probability. (Here $\tilde O$ hides some polylog factors.) Since $T$ is typically at least linear in n, this means that the additional $1/\sqrt{n}$ term will not be dominating.
>
> On the other hand, for our new bound, which is derived from Bayes-Stability instead of uniform stability, it remains unknown whether it can be translated into a high-probability bound following a similar approach. We believe that it is an interesting open problem to prove a similar high-probability bound (with a small overhead) for the Bayes-Stability framework.
>
> We will include the above discussion into the next version.
>
> Indeed, [Rivasplata et al.] also combines ideas from PAC-Bayes and stability. While their stability is actually the hypothesis stability measured by the distance on the hypothesis space. And their work stduies the case when the returned hypothesis (model parameter) is randomized by a Gaussian perturbation (i.e., the posterior $Q = \mathcal{N}(A(S),\sigma I)$), which is different from our work. Thanks for pointing this out. In the next version, we will discuss their work and modify our descriptions in the introduction and abstract.
>
> -------------------------------------
> Reference
>
> PAC-Bayes bounds for stable algorithms with instance-dependent priors. Rivasplata et al.
>
> High probability generalization bounds for uniformly stable algorithms with nearly optimal rate. Vitaly Feldman and Jan Vondrak.
>
> Sharper bounds for uniformly stable algorithms. Bousquet et al.
>
> Generalization bounds for uniformly stable algorithms. Feldman et al.

---

### Decision · Program_Chairs · 2019-12-19

**Decision:**

Accept (Poster)

**Comment:**

The authors provide bounds on the expected generalization error for noisy gradient methods (such as SGLD). They do so using the information theoretic framework initiated by Russo and Zou, where the expected generalization error is controlled by the mutual information between the weights and the training data. The work builds on the approach pioneered by Pensia, Jog, and Loh, who proposed to bound the mutual information for noisy gradient methods in a step wise fashion.

The main innovation of this work is that they do not implicitly condition on the minibatch sequence when bounding the mutual information. Instead, this uncertainty manifests as a mixture of gaussians. Essentially they avoid the looseness implied by an application of Jensen's inequality that they have shown was unnecessary.

I think this is an interesting contribution and worth publishing. It contributes to a rapidly progressing literature on generalization bounds for SGLD that are becoming increasingly tight.

I have one strong request that I will make of the authors, and I'll be quite disappointed if it is not executed faithfully.

1. The stepsize constraint and its violation in the experimental work is currently buried in the appendix. This fact must be brought into the main paper and made transparent to readers, otherwise it will pervert empirical comparisons and mask progress.

2. In fact, I would like the authors to re-run their experiments in a way that guarantees that the bounds are applicable. One approach is outline by the authors: the Lipschitz constant can be replaced by a max_i bound on the running squared gradient norms, and then gradient clipping can be used to guarantee that the step-size constraint is met.  The authors might compare step sizes, allowing them to use less severe gradient clipping. The point of this exercise is to verify that the learning dynamics don't change when the bound conditions are met. If they change, it may upset the empirical phenomena they are trying to study. If this change does upset the empirical findings, then the authors should present both, and clearly explain that the bound is not strictly speaking known to be valid in one of the cases. It will be a good open problem.

---

> ### Author Response · Authors · 2020-02-11
> **Update**
>
> Thanks for your comments! We have addressed your two concerns in our new version (see Appendix C.2).